# Scaling Policy Compliance Assessment in Language Models with Policy Reasoning Traces

## Abstract

Policy compliance assessment is a fundamental task of evaluating whether an input case strictly complies with a set of human-defined rules, more generally known as *policies*. In practice, human experts follow a systematic, step-by-step process to identify violations with respect to specific stipulations outlined in the policy. However, such documentation of gold-standard, expert-level reasoning processes is costly to acquire. In this paper, we introduce POLICY REASONING TRACES (PRT), a form of specialized generated reasoning chains that serve as a *reasoning bridge* to improve an LLM's policy compliance assessment capabilities. Our empirical evaluations demonstrate that the use of PRTs for both inference-time and training-time scenarios significantly enhances the performance of open-weight and commercial models, setting a new state-of-the-art for HIPAA and GDPR policies. Beyond accuracy gains, we also highlight how PRTs can improve an LLM's ability to accurately cite policy clauses, as well as influence compliance decisions through their high utilization from the raw chains-of-thought.

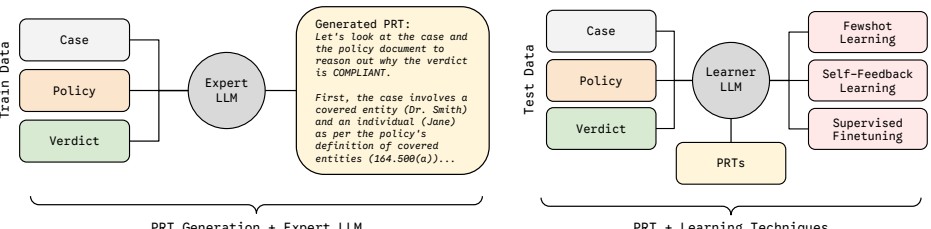

Figure 1: POLICY REASONING TRACES (PRTs) are derived from querying a frontier *pseudo-expert* (e.g., DEEPSEEK-R1) reasoning model using datasets of cases and gold-standard verdicts with respect to a policy. Generated PRTs serve as a *reasoning bridge* that connects policy-specific constraints and nuances to compliance judgments, which can be used off-the-shelf as in-context demonstrations or as a compilation for SFT to improve the compliance assessment capabilities of models.

## 1 Introduction

When a court examines if a medical transaction is compliant with data privacy regulations (e.g., GDPR), it conducts a systematic examination of which provisions from the law have been violated and issues a corresponding verdict. Automating this process presents a broader, more fundamental challenge of *policy compliance assessment*, where large language models (LLMs) are expected to correctly assess if a given case scenario fully complies to a set of human-defined rules—more generally known as *policies*[1]. In addition, learning to make determinations that align with a domain expert's judgment and correctly apply the stipulations of a policy before providing a final verdict is a critical capability for an LLM to develop, especially for high-stakes domains such as healthcare, education, and security (Chen et al., 2025; Imperial et al., 2025). The main challenge in delegating

---

[1]Not to be confused with the term *policy* in reinforcement learning, where this denotes the learned mapping of states to actions.

policy compliance assessment to LLMs is that interpreting policies requires expert-level knowledge to identify which constraints and provisions are applicable. While domain experts acquire this form of knowledge through subject matter expertise and accumulation of experience (Ruan et al., 2025; Phan et al., 2025; Bedi et al., 2025; Malaviya et al., 2025; Guha et al., 2023), asking humans to record every detailed logical reasoning for each stipulation of a policy is extremely expensive and time-consuming.

In this paper, we aim to *bridge* the lack of gold-standard systematic reasoning traces from domain experts. We propose the use of POLICY REASONING TRACES (PRT), a novel approach that aims to improve the policy compliance capabilities of state-of-the-art LLMs. As illustrated in Figure 1, PRTs are demonstrations of the *pseudo-expert* reasoning process generated from querying frontier reasoning LLMs using policy compliance datasets that only provide case information and gold-standard verdicts with respect to a policy. The use of PRTs is intuitive: they serve as a *reasoning bridge* that links policy-specific constraints and nuances to compliance judgments, which can be used off-the-shelf as in-context learning (ICL) via few-shot demonstrations or compiled into datasets for supervised finetuning (SFT). By integrating PRTs into the learning process, we move towards a more intuitive approach in tackling policy compliance assessment, where expert-like rationales are used instead of reducing the task to a simple verdict prediction.

To empirically investigate the effectiveness of PRTs, we evaluate them on three diverse multi-domain policies, including HIPAA and GDPR for healthcare and general data privacy, as well as OpenAI's ModelSpec for model interaction safety. We compare performances using few-shot in-context learning and self-feedback with added PRTs across a variety of frontier open-weight and commercial reasoning LLMs, including DEEPSEEK-R1, GPT-5-MINI, and QWEN2.5-7B to name a few. We demonstrate that using PRTs as in-context demonstrations enables open-weight LLMs to achieve a 50-100% significant performance increase for HIPAA and sets new state-of-the-art baselines for GDPR through finetuning. Additional experiments on cross-policy generalization reveal that PRTs enable LLMs to transfer learned compliance assessment knowledge across domains (especially from HIPAA to ModelSpec and GDPR to HIPAA) as well as boost the ability of LLMs to cite the correct policy clauses when reasoning towards a verdict.

## 2 POLICY REASONING TRACES (PRT)

### 2.1 MOTIVATION

Our primary motivation for conceptualizing policy reasoning traces (PRTs) stems from the idea that automating policy-dependent tasks can primarily benefit from a resource of available reference examples that demonstrate policy-based reasoning. This scenario reflects the actual procedures in the legal domain, where courts refer to outcomes from case precedents (previous case decisions) and identify specific provisions of a policy that are applicable before issuing a verdict and often emulated in legal NLP works (Yang et al., 2025b; Chen & Zhang, 2025; Liu et al., 2025a; Fan et al., 2024). In the context of this study, we hypothesize that LLMs can benefit from generated PRTs through either inference-time or training-time integrations to make accurate predictions and improve their performance in policy compliance assessments.

We sketch our formalization of PRTs and how these grounded policy-based reasoning augmentations are generated and used. We take a high-level approach in our formalization, as this concept can also be applied to other policy-dependent domains.

### 2.2 TASK FORMALIZATION

Let $\mathcal{P}$ denote a policy document composed of a collection of written constraints or rules $r$. Each rule may optionally be associated with one or more gold-standard reference examples $e$, each annotated with a verdict $v$ whether it is COMPLIANT or NON-COMPLIANT relative to the rule. We formalize this as follows:

$$\mathcal{P} = \left\{ \left( r_i, \{e_{ij}, v_{ij}\}_{j=1}^{n} \right) \right\}_{i=1}^{m} \tag{1}$$

Given this, we introduce the fundamental task of *policy compliance assessment*, which makes use of a predictive model $\mathcal{M}$ to predict a single binary verdict $v$ whether an input case $c$ is COMPLIANT or NON-COMPLIANT with a given policy $\mathcal{P}$. We assume that $\mathcal{M}$ is a reasoning model that can produce a reasoning trace or intermediary tokens first before providing a final verdict. We formalize this process as follows:

$$\mathcal{M}(c, \mathcal{P}) = v, \quad v \in \{\text{COMPLIANT, NON-COMPLIANT}\}. \tag{2}$$

## 2.3 PRT Generation

We start from a given dataset of existing case-verdict pairs $(c_i, v_i)$ where the cases are documented narrative scenarios and their associated verdicts $v \in \{\text{COMPLIANT, NON-COMPLIANT}\}$ based on the policy $\mathcal{P}$ are considered gold-standard. We treat this as the train data $\mathcal{D}_{\text{train}}^{\mathcal{P}}$ where PRTs will be generated from using an expert reasoning model $\mathcal{M}_{\text{E}}$. For each instance $(c_i, v_i)$, an expert reasoning model $\mathcal{M}_{\text{E}}$ is used to generate a corresponding PRT:

$$\text{PRT}_i = \mathcal{M}_{\text{E}}(c_i, \ \mathcal{P}, \ v_i). \tag{3}$$

The resulting augmented train data with PRTs dataset is now:

$$\mathcal{D}_{\text{train}}^{\mathcal{P},\text{PRT}} = \{(c_i, v_i, \text{PRT}_i)\}_{i=1}^{n}. \tag{4}$$

By providing gold-standard case-verdict pairs $(c_i, v_i)$, we can assure that the PRT traces generated are grounded on information that ties the reasoning to the gold-standard information when resolving nuanced constraints from the policy.

## 2.4 Inference and Finetuning with PRTs

For inferring the policy compliance of a new unseen test case, we apply the same principles described earlier. Given a new input case $c^*$, a sample of PRTs selected from the PRT-augmented train data $\mathcal{D}_{\text{train}}^{\mathcal{P},\text{PRT}}$, and the same policy $\mathcal{P}$ where the PRTs were generated, a learner reasoning model $\mathcal{M}_{\text{L}}$ is used to predict the most-applicable verdict $v^*$. We formalize this learning process as follows:

$$v^* = \mathcal{M}_{\text{L}}(c^*, \ \mathcal{P}, \ \text{PRT}) \tag{5}$$

Depending on the learning paradigm, $\mathcal{M}_{\text{L}}$ can be optimized through various learning techniques such as in-context learning, feedback learning, and imitation learning using the PRTs from $\mathcal{D}_{\text{train}}^{\mathcal{P},\text{PRT}}$ to improve its compliance capabilities further.

For **few-shot in-context learning (ICL)**, $\mathcal{M}_{\text{L}}$ can be conditioned based on a selected subset of reference case demonstrations with gold-standard verdicts and corresponding PRTs from the train data $\{(c_j, v_j, \text{PRT}_j)\}_{j=1}^{k} \subset \mathcal{D}_{\text{train}}^{\mathcal{P},\text{PRT}}$ and predicts:

$$v^* = \mathcal{M}_{\text{L}}\big(c^*, \mathcal{P}, \ \{(c_j, v_j, \text{PRT}_j)\}_{j=1}^{k}\big). \tag{6}$$

Likewise, **supervised finetuning (SFT)** can be done on $\mathcal{M}_{\text{L}}$ using the compiled PRT-augmented train data $\mathcal{D}_{\text{train}}^{\mathcal{P},\text{PRT}}$ by minimizing cross-entropy loss over the given gold-standard verdicts $v_i$. However, unlike with ICL, we extract only the relevant policy clause information for each case $\mathcal{P}_i \subseteq \mathcal{P}$ in order not to overload the context length of $\mathcal{M}_{\text{L}}$. Hence, the process goes:

$$\theta_{\text{L}}^* = \arg\min_{\theta_{\text{L}}} \ \mathcal{L}(\theta_{\text{L}}), \qquad \mathcal{L}(\theta_{\text{L}}) = -\sum_i \log p_\theta(v_i \mid c_i, \text{PRT}_i, \mathcal{P}_i), \tag{7}$$

Once optimized, inference[2] with new cases $c^*$ can be done by conditioning on the case information, policy text, and corresponding PRTs under the same policy $\mathcal{P}$ to predict the verdict as shown below:

---

[2]Note that at inference time, $\mathcal{M}_{\text{L}}$ does not receive a PRT for $c^*$, only the PRTs exclusively from $\mathcal{D}_{\text{train}}^{\mathcal{P},\text{PRT}}$. This prevents $\mathcal{M}_{\text{L}}$ from simply copying or extracting the verdict from any source and requires it to generalize the reasoning patterns to produce a verdict.

$$v^* = \mathcal{M}_{\mathrm{L}}(c^*, \mathrm{PRT}, \mathcal{P}; \theta_{\mathrm{L}}^*). \tag{8}$$

Both learning paradigms operationalize the learned compliance capabilities of $\mathcal{M}_{\mathrm{L}}$ to reason and predict the most applicable verdict for new, unseen cases.

## 3 EXPERIMENT SETUP

**Policies and Test Datasets.** Our main criteria for selecting a dataset to be included in the evaluation are that the policy text should be publicly available and that the dataset contains case demonstrations and expert labels based on their compliance with the policy. Overall, we have come up with the following policy compliance datasets that span across the domains of healthcare, data privacy, and model safety interactions to be used in our experiments:

- **Health Insurance Portability and Accountability Act (HIPAA)**[3]. The HIPAA Privacy Rule stipulates the use and disclosure of covered entities' protected health information (PHI), including individuals and organizations. We specifically use Subpart E (Privacy of Individually Identifiable Health Information) of HIPAA from 164.500 to 164.530, which establishes around 15 sectional provisions regarding who and what requirements apply and totals $\approx$3.9K tokens. As a corresponding test dataset, we use the **GOLDCOIN-HIPAA** train and test sets for compliance assessment used by Fan et al. (2024), which contain 309 and 107 synthetic court cases with compliance verdicts quality-checked by legal experts, respectively.

- **General Data Protection Regulation (GDPR)**[4]. Similar to HIPAA, GDPR is a comprehensive data privacy law that aims to regulate the collection of personal data from residents of the European Union and how organizations handle and process this data lawfully and securely. For GDPR, we use Articles 1 to 90, which totals to $\approx$8.8K tokens and covers foundational aspects of data protection, rights of data subjects, responsibilities of controllers and processors, and data transfers, to name a few. To build the right dataset for our task, we requested data from **GDPRHub**[5], a public repository of GDPR-related court cases from Data Protection Authorities (DPAs) across Europe. We compiled 764 and 326 real-world court cases, with gold-standard legal reasoning and compliance verdicts, for our train and test sets, respectively.

- **OpenAI Model Specifications (ModelSpec)**[6]. The ModelSpec is an extensive policy specification that outlines guidance for desired safe and harmless model behavior used by OpenAI for their LLM products (e.g., ChatGPT), applicable to both chat interfaces and APIs. We utilize all 20 sections of ModelSpec, which totals to $\approx$4.1K tokens. To build the train data, we use 64 examples of user interactions tagged by ModelSpec as good (safety compliant) or bad (potentially harmful). For the test set, we use **XSTest** (Röttger et al., 2024), which contains 450 mixed instances for measuring exaggerated safety in LLMs. We justify the use of XSTest in this work for measuring compliance to model safety policies, given its wide usage for the same purpose in previous works (Guan et al., 2024; Chao et al., 2024; Han et al., 2024; Röttger et al., 2024)

With respect to the task, all datasets associated with each policy have instances labelled with either COMPLIANT or NONCOMPLIANT tags. There are no overlaps between the train and test sets for each policy that may cause contamination. We perform style reformatting and minor summarization using GPT-5-MINI for all policies to standardize the policy text for prompting and finetuning setups in the experiments. More information on this process is covered in Appendix I.

**Expert Models for PRT Generation.** For selecting expert models to generate PRTs, our main rule-of-thumb is that the models should be considered performant across reasoning-based tasks and have been trained on significant domain-specific data. Hence, we select two types of expert models that will generate two forms of PRTs:

---

[3]https://www.hhs.gov/hipaa/for-professionals/privacy/laws-regulations/index.html
[4]https://gdpr-info.eu/
[5]https://gdprhub.eu/
[6]https://model-spec.openai.com/2025-04-11.html

**INPUT CASE INFORMATION** (FAN ET AL., 2024)

Jane's doctor, Dr. Smith, discovered she had a rare, reportable infectious disease during a checkup. By law, he reported the case to the state health department, providing her personal and medical details to help track and control the outbreak. Jane's consent was not required since disclosure was legally mandated. The health department confirmed receipt and assured the information would be used solely for public health purposes.

| *Generalist* PRT (DEEPSEEK-R1) | *Specialist* PRT (SAULLM-54B) |
|---|---|
| 1. The policy states compliant content SHOULD allow disclosures required by law or for public health activities and permits disclosures for public health activities like disease control. 
 2. The case matches these allowances. Looking at Section 164.512(a), the disclosure was required by law and for public health activities, permitted without authorization; the health department is an appropriate recipient; no prohibited uses occurred... 
 10. Therefore, the case is COMPLIANT. | 1. The case involves a covered entity (Dr. Smith) and an individual (Jane) as covered entities (164.500(a)). 
 2. The case describes a covered entity (Dr. Smith) contacted the state health department to report a case of a rare infectious disease as required by law (164.512(a)). 
 3. The policy explicitly states that covered entities may disclose protected health information (PHI) as required by law (164.512(a))... 
 10. Therefore, the case is COMPLIANT. |

Figure 2: Selected examples of portions of generated *Generalist* and *Specialist* PRTs for HIPAA and an instance of input case from the training data. Upon manual inspection, PRT generated from the *Specialist* model SAULLM-54B tend to be more frequent in citing policies, while the *Generalist* ones are more conversational and thinking-like. We provide additional details in manually inspecting PRTs in Appendix C.

- *Generalist* **Model**. For this type of expert reasoning model, we look for all-around high performance across multipurpose reasoning benchmarks that have not been trained or optimized for a specific domain. For this, we select DEEPSEEK-R1 (Guo et al., 2025) with 37B active parameters (671B total) due to its recognition as a general state-of-the-art frontier reasoning model, as well as allowing access to its raw reasoning traces for constructing PRTs via API.

- *Specialist* **Model**. For this type of expert reasoning model, we require specific pretraining, instruction-tuning, and optimization to an overlapping domain related to HIPAA, GDPR, and ModelSpec. We also require the model to be of substantial size to be comparable to the *Generalist* model. Hence, we select SAULLM-INSTRUCT with 46.7B active parameters (54B total), which has been pretrained and instruction-tuned with massive legal data spanning 520B tokens and beats GPT-4 on legal benchmarks (Colombo et al., 2024; Guha et al., 2023).

For generating the PRTs for in-context demonstrations and SFT experiments, we use the train splits with gold-standard verdicts of COMPLIANT or NON-COMPLIANT from the associated policy datasets for HIPAA, GDPR, and ModelSpec and queried both *Generalist* and *Specialist* models. The prompts used in this process and more information can be found in the Appendix B and I.

**Learner Models and Assessment Methods.** We evaluate the policy compliance capabilities assessment using a representative set of open-weight and commercial reasoning LLMs. Specifically, we experiment with DEEPSEEK-R1 and DEEPSEEK-R1-LLAMA-8B (Guo et al., 2025), GEMINI-2.5-FLASH (Comanici et al., 2025), QWEN2.5-7B and QWEN2.5-32B (Yang et al., 2025a), GPT-5-MINI and GPT-OSS (Agarwal et al., 2025). We use these LLMs for the two learning paradigms for policy compliance assessment, specifically inference-time assessment through **in-context learning (ICL)** via few-shot demonstrations and training-time assessment via **supervised finetuning (SFT)**.

In terms of methods of querying the LLM to produce assessments, we explore the following widely recognized prompt-based setup:

- **Standard Prompting.** This is the simplest, most basic setup of prompting a model for evaluating policy compliance. We define BASE and FEW-SHOT as two forms of standard prompting. For BASE, we only provide the input case $c^*$ being evaluated and policy text $\mathcal{P}$ as sources of information, while for FEW-SHOT, we additionally provide randomly selected case-verdict $(c, v) \in \mathcal{D}_{\text{train}}^{\mathcal{P}}$ pairs without PRTs for assessment as with conventional practice in few-shot in-context learning (Brown et al., 2020).

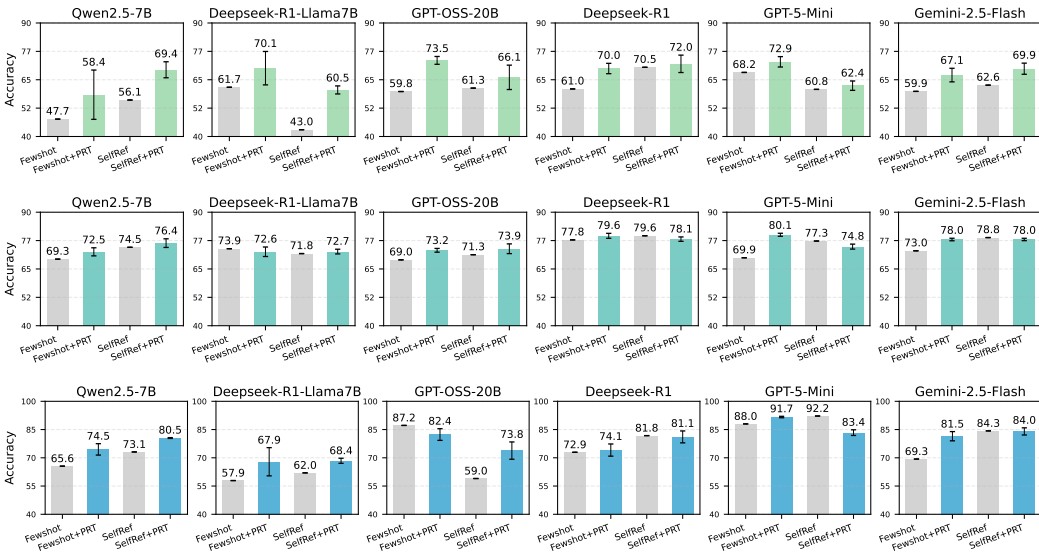

Figure 3: Inference-time policy compliance using few-shot and self feedback-based approaches. We aggregate the mean performances of using *Generalist* and *Specialist* PRTs across state-of-the-art open-weight and commercial models. We evaluate three diverse policies, including HIPAA (**top**) and GDPR (**middle**) for healthcare and general data privacy, and ModelSpec (**bottom**) for model interaction safety. The full table of performance can be found in Appendix A.

- **Self Feedback.** This is an advanced version of STANDARD PROMPTING where the process of assessment allows the model to reflect over its reasoning first through self-feedback or refinement via SELF-REFINE (Madaan et al., 2023) before providing a final judgment. We use only one round of SELF-REFINE for fair comparison and practicality with our compute budget.

- **(Method) + PRTs.** This setup augments three (3) instances of case, verdict, and PRT $\{c, v, \text{PRT}\} \in \mathcal{D}_{\text{train}}^{\mathcal{P}, \text{PRT}}$ demonstrations from the training data picked through random selection (RAND) or most similar (REL) to the two previous setups mentioned. For selecting the most similar $\{c, v, \text{PRT}\}$ triples, we prompt GPT-5-MINI to compare the information from the input case and provide the three (3) most similar case instances from $\mathcal{D}_{\text{train}}^{\mathcal{P}, \text{PRT}}$.

Due to the nature of the task, we select models that can handle at least 8192 tokens for context length to fully process the entirety of the policy texts and PRTs as inputs. We explicitly state in our prompts for the setups discussed that the models should reason first before giving the final verdict. The full details of our experiment, including hyperparameters, configurations, and prompts for each method, can be found in the Appendix D and I.

## 4 RESULTS

### 4.1 INFERENCE-TIME AND TRAINING TIME POLICY COMPLIANCE ASSESSMENT

**PRTs Improve Policy Compliance Assessment of Open-Weight and Commercial Models.** As reported in Table 4 and visualized in Figure 3, we observe that the addition of PRTs as few-shots in prompts significantly improves[7] the performance of reasoning models for the HIPAA policy, gaining as large as 16-30 points boost in accuracy (more than 50%) for open-weight models such as QWEN2.5-7B and DEEPSEEK-R1-LLAMA-8B. Likewise, we see the same performance upgrade in commercial reasoning models such as GPT-5-MINI and GEMINI-2.5-FLASH with 5-16 raw point improvement using PRTs in-context. For GDPR, we achieve a new state-of-the-art performance with

---

[7]For HIPAA, conducting one-sided paired $t$-tests (random and relevant PRTs vs. no PRTs) results to a *significance* with corrected $p$-values of $p = 0.0005$ and $p = 0.002$ under Bonferroni correction, and $p = 0.0005$ and $p = 0.0012$ under Holm correction.

| SETUP | MODEL | ACC |
|---|---|---|
| *HIPAA* | | |
| Previous SOTA | GOLDCOIN (Fan et al., 2024) | 79.9 |
| Best Baseline (no PRT) | DEEPSEEK-R1 | 70.5 |
| Best Few-shot + PRT | DEEPSEEK-R1 | 77.7 |
| Best SFT + PRT | QWEN2.5-7B | **81.3** |
| *GDPR* | | |
| Previous SOTA | - | - |
| Best Baseline (no PRT) | GEMINI-2.5-FLASH | 79.5 |
| Best Few-shot+PRT | DEEPSEEK-R1 | 81.0 |
| Best SFT+PRT | QWEN2.5-7B | 78.8 |
| *ModelSpec* | | |
| Previous SOTA | DELIBERATIVE ALIGNMENT (Guan et al., 2024) | **93.0** |
| Best Baseline (no PRT) | GPT-5-MINI | 92.7 |
| Best Few-shot+PRT | GEMINI-2.5-FLASH | 86.6 |
| Best SFT+PRT | QWEN2.5-7B | 86.2 |

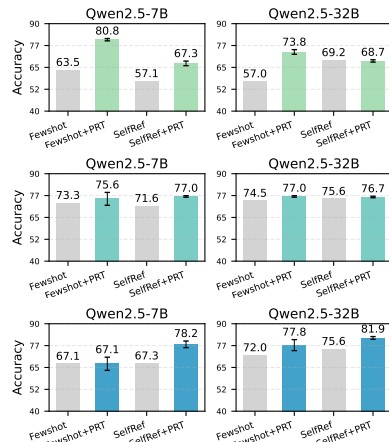

Figure 4: **Left**: Comparison of best-performing baseline models, PRT-enhanced models, and state-of-the-art methods for each policy. Using PRTs beats previous work's performance on HIPAA by up to +1.4% while also boosting open-weight model performance to be comparable to optimized commercial models such as O1 with <10% difference. **Right**: Finetuned QWEN-2.5-7B and 32B instruct models also exhibit benefits in performance when using PRTs via few-shot or self feedback. Bar graphs arranged in the order from top to bottom: HIPAA, GDPR, and ModelSpec.

accuracies of 81.0 using DEEPSEEK-R1 and GPT-5-MINI with *Specialist* and *Generalist* PRTs, respectively. This improvement is significant[8] and echoes the similar advantage of PRTs with HIPAA as few-shot reasoning guides that work to scale the performance of commercial frontier models further. Across the board, we observed the majority advantage of using *Generalist* over *Specialist* PRTs and used this form of PRTs for the succeeding closer-look experiments.

**PRTs Sets New SOTA for Legal Policy Compliance Assessment.** We compare the results of state-of-the-art methods with our implementation of finetuning[9] on PRTs for reasoning models, including QWEN2.5-7B and 32B models, as well as the distilled LLAMA version of DEEPSEEK-R1 in Table 5. On HIPAA, finetuning on PRTs achieves 80-81% accuracy and beats state-of-the-art methods like GOLDCOIN (Fan et al., 2024), which is anchored on contextual integrity theory tailored for HIPAA-specific elements (Nissenbaum, 2004). Likewise, using PRTs as few-shot demonstrations helps models such as DEEPSEEK-R1-LLAMA-8B reach a performance close to GOLDCOIN-optimized models, 77.7% against 79.9%. As mentioned earlier, using PRTs also sets the state-of-the-art baseline for GDPR with 81.0% accuracy using DEEPSEEK-R1. We also note an *equalizing effect* of PRTs, which helps boost the performances of open-weight models (e.g., QWEN2.5-7B) to reach commercial ones (e.g., GPT-5-MINI) as seen on the best setup comparisons table in Figure 4.

### 4.2 INTERACTIONS ON SAFETY OPTIMIZATIONS AND CROSS-POLICY GENERALIZATION

**PRTs May Provide Limited Gains for Doubly-Policy Optimized Models.** We observe declines in performance, averaging 4.7 in accuracy, upon using PRTs with OpenAI models GPT-OSS-20B and GPT-5-MINI for ModelSpec. We posit that this occurs with models we consider *doubly-optimized* using the same policy for the task of policy compliance assessment. Upon cross-checking literature, we find that OpenAI is natively optimizing models such as O-series, GPT-4, and GPT-5 with RL alignment techniques such as DELIBERATIVE ALIGNMENT with an expanded version of ModelSpec, and possibly larger, in-house safety benchmarks (Guan et al., 2024). Hence, these models perform

---

[8]For GDPR, conducting one-sided paired $t$-tests (random and relevant PRTs vs. no PRTs) results to a *significance* with corrected $p$-values of $p = 0.0289$ and $p = 0.0396$ under Bonferroni correction, and $p = 0.0289$ and $p = 0.0289$ under Holm correction.

[9]Conducting Cohen's $d$ effect sizes on the performance of finetuned QWEN2.5-7B and DEEPSEEK-LLAMA8B against non-finetuned models shows medium to large positive gains for HIPAA ($d = 0.674$ and $d = 0.895$ using zero-shot and few-shot settings) and extremely large positive gains for GDPR ($d = 0.631$ and $d = 4.602$ for the same settings as HIPAA).

| Setup | MODELSPEC | DYNABENCH |
|---|---|---|
| LlamaGuard3-8B | 59.3 | 13.1 |
| WildGuard-7B | 45.1 | 20.9 |
| GuardReasoner-8B | 44.2 | 22.0 |
| NemoGuard-8B | 78.8 | 23.7 |
| ShieldGemma-9B | 55.5 | 38.2 |
| DynaGuard-8B | 85.8 | 72.5 |
| Qwen2.5-7B (Fewshot + PRT) | 80.7 | 61.1 |
| Qwen2.5-7B (SFT + PRT) | 86.2 | 56.9 |
| Qwen2.5-32B (SFT + PRT) | 82.4 | 79.0 |
| Gemini-2.5-Flash (Fewshot + PRT) | 83.3 | 94.1 |
| GPT-5-Mini (Fewshot + PRT) | 92.1 | 95.5 |

Table 1: Comparison of PRT-enhanced models against external safety-optimized guard models on ModelSpec compliance and generalization with user-defined multidomain safety policies through DYNABENCH (Hoover et al., 2025). Using PRTs enables both open-weight and commercial models to effectively generalize their compliance capabilities to out-of-domain unique policies and even outperform most guard models (e.g., DYNAGUARD-8B finetuned on a similar DYNABENCH task distribution).

better using standard prompts and yield higher results than using PRTs, and they also avoid the overthinking phenomenon (Gema et al., 2025; Sui et al., 2025). Counter to this, other commercial reasoning models like GEMINI-2.5-FLASH that do not specifically use ModelSpec exhibit significant positive gains using PRTs as in-context demonstrations, averaging 13.0 increase in accuracy.

**Models Finetuned on PRTs Exhibit Strong Policy Generalization.** We visualize the results of our cross-policy domain generalization experiments in Figure 13. For this study, we finetune QWEN2.7-7B-INSTRUCT on each policy's PRT train dataset and test them on each policy's test data while exploring setups with and without PRTs in-context. We observe that QWEN2.7-7B-INSTRUCT finetuned on HIPAA PRT data generalizes well to GDPR and ModelSpec test data, achieving high accuracies with 78.5 and 86.6, respectively. To compare, GPT-4O optimized via DELIBERATIVE ALIGNMENT (Guan et al., 2024) obtains 88.0 on the same test set, which further supports the assistance of finetuning PRTs for better generalization. Similar to using PRTs as few-shots reported in Table 4, all models finetuned on each of the three policies separately gain doubled policy generalization performance, from 45.8 to 78.5 (+32.7) and 74.8 (+29) using GDPR data and from 36.4 to 77.6 (+41.2) and 74.3 (+37.9) using ModelSpec data, upon evaluating on the HIPAA test data when PRTs are used. Conducting paired $t$-test on within-policy and cross-policy values results in a non-significance[10] in performance difference while Cohen's $d$ reveals negligible effect size ($d = 0.04$) between the two groups, which suggests strong policy generalization.

**PRT-Enhanced Models Outperform and Outgeneralize External Guard Models on Dynamic Policies.** We compare the performance of our best-performing PRT-enhanced models through few-shot in-context learning and finetuning with external safety-optimized guard models, specifically LLAMAGUARD3-8B (Grattafiori et al., 2024), WILDGUARD-7B (Han et al., 2024), NEMOGUARD-8B (Ghosh et al., 2025), GUARDREASONER-8B (Liu et al., 2025b), SHIELDGEMMA-9B (Zeng et al., 2024), and DYNAGUARD-8B (Hoover et al., 2025). In terms of ModelSpec compliance through XSTest, we outperform most external guard models with our PRT approach with GPT-5-MINI obtaining the top score of 92.1. Finetuning QWEN2.5-7B with PRTs also beats the best guard model DYNAGUARD-8B despite only being conditioned on 64 PRT demonstrations. We also evaluate the same set of models with DYNABENCH (Hoover et al., 2025), a new benchmark for assessing compliance with user-defined multidomain safety policies. Results presented in Table 1 further support the strong generalization capabilities provided by PRTs, as evidenced by the PRT-enhanced commercial and open-weight models outperforming the guard models on out-of-domain policies via DYNABENCH.

---

[10]Within-policy mean accuracy = 0.668, cross-policy mean accuracy = 0.695; $t$ = –0.146, $p$ = 0.886.

| MODEL | POLICY | $\mu_{\text{cited}}$ | RECALL (%) | | | EXACT-MATCH (%) | | | Top Incorrect Clause |
|---|---|---|---|---|---|---|---|---|---|
| | | | No-PRT | PRT | Δ | No-PRT | PRT | Δ | |
| QWEN2.5-7B (FEW-SHOT) | HIPAA | 5.17 | 18.2 | 20.1 | +1.9 | 0.0 | 0.0 | - | Section 164.502(b) |
| | GDPR | 6.81 | 49.5 | 59.0 | +9.5 | 5.8 | 0.9 | -4.9 | Article 5 (Principles in processing personal data) |
| | ModelSpec | 5.20 | 28.5 | 42.2 | +13.7 | 0.0 | 0.2 | +0.2 | Respect the spirit of instructions. |
| QWEN2.5-7B (SFT) | HIPAA | 5.17 | 16.8 | 26.3 | +9.5 | 0.9 | 0.0 | -0.9 | Section 164.502(b) |
| | GDPR | 5.99 | 46.2 | 59.3 | +13.1 | 7.1 | 1.8 | -5.2 | Article 6 (Lawfulness of processing) |
| | ModelSpec | 5.70 | 31.8 | 44.4 | +12.6 | 0.0 | 0.2 | +0.2 | Stay in bounds. |
| GPT-5-MINI | HIPAA | 8.60 | 37.7 | 43.9 | +6.1 | 0.0 | 0.0 | - | Section 164.502(b) |
| | GDPR | 13.39 | 77.0 | 86.2 | +9.3 | 2.1 | 0.0 | -2.1 | Article 25 (Data protection by default) |
| | ModelSpec | 5.21 | 33.1 | 63.1 | +30.0 | 0.0 | 8.7 | +8.7 | Do not encourage illicit behavior. |

Table 2: Policy clause relevance result of finetuned QWEN2.5-7B-INSTRUCT models compared with state-of-the-art commercial model GPT-5-MINI. The addition of PRTs enables models to incorporate the correct policy clauses into their reasoning, based on high RECALL scores, thereby improving their practical usability. The Δs denote changes in values from No-PRT to using PRT.

### 4.3 POLICY CLAUSE RELEVANCE AND REASONING PERSISTENCE OF PRTs

**PRTs Allow Models to Scope Relevant Policy Clauses.** Benchmarks evaluating legal reasoning of LLMs across retrieval-based tasks (e.g., clause retrieval) frequently emphasize the importance of LLMs to correctly identify the applicable clauses as the basis of predictions to have value in real-world legal applications (Liu et al., 2025a; Chalkidis et al., 2022; Guha et al., 2023; Wu et al., 2020). We measure how QWEN2.5-7B using PRTs as few-shot in-context demonstrations and a version finetuned on PRTs affects their ability to cite the correct policy clauses. We also compare GPT-5-MINI as the closest available frontier commercial LLM. Results reported in Table 2 clearly show the substantial advantage of using PRTs to allow models to scope the correct policy clauses in their reasoning, as evidenced by high positive gains on RECALL scores (+2.0-9.0). For EXACT-MATCH, which penalizes under- and over-reference, small deteriorations are expected given that PRTs are used as examples, and this allows models to cite more policy clauses in their reasoning.

**PRTs Have High Utility Rates from Raw Chain-of-Thought.** Monitoring the raw chain-of-thought provides an opportunity to analyze how LLMs solve complex, high-risk tasks by diagnosing reasoning steps that might not be visible from the output alone (Korbak et al., 2025; Barez et al., 2025; Chen et al., 2025). To gain a real measure of the true utilization of PRTs, we analyze DEEPSEEK-R1's raw chain-of-thought reasoning traces only available through the official API[11] when using PRTs as few-shot in-context demonstrations. To automatically flag notions of mentioning PRTs in the raw CoT texts, we prompt GPT-5-MINI to identify phrases such as *"Based on the example reasoning..."* or *"Looking at case, verdict, PRT examples..."* for each instance from the policy test datasets (prompt details in Appendix I). Results reported in Table 3 show that using PRTs as few-shot demonstrations achieves a high utility rate of 80% and above across HIPAA, GDPR, and ModelSpec for assessing policy compliance. We also observe that the mean reference value ($\mu_{\text{ref}}$), which denotes the frequency of reference the model makes to the given PRT demonstrations, is more prominent in safety policies like ModelSpec (6.0-7.2) than HIPAA and GDPR (1.2-1.9) and directly proportional to utility percentage. We find these results as a strong evidence in models fully utilizing PRTs as a reasoning bridge to provide confident policy compliance judgments.

## 5 RELATED WORK

**AI for Constitutional and Policy Compliance.** Early explorations of transforming policy texts, such as regulations in the legal domain, used simple knowledge representations like logic formalization programs to assist potential integration to rule-based AI (Sergot et al., 1986; Kowalski, 1992). This was further extended by Bench-Capon et al. (1991) and Bench-Capon et al. (1993) to provide explanations to the initial logic programs as a form of justification before giving a final judgment. We consider this the earliest form of *reasoning bridge*, closely related to our motivation behind PRTs. Current works now mainly use ML models paired with advanced knowledge processing techniques

---

[11] https://api-docs.deepseek.com/

such as deep learning and retrieval architectures (Sun et al., 2025; Zoubi et al., 2024; Mousavi Nejad et al., 2020), task decomposition (Saeidi et al., 2021), and anchors to privacy and policy frameworks (Li et al., 2025; Fan et al., 2024; Hassani et al., 2024). State-of-the-art advancements in LLMs through post-training techniques like instruction-tuning (Chung et al., 2024; Longpre et al., 2023; Wei et al., 2022a), chain-of-thought prompting (Wei et al., 2022b), and preference optimization (Rafailov et al., 2024; Ouyang et al., 2022; Christiano et al., 2017) allowed LLMs to gain even better generalization capabilities across diverse tasks, including policy compliance (Kumarage et al., 2025; Han et al., 2025; Bolton et al., 2025; Masoudifard et al., 2024; Imperial et al., 2024; Mu et al., 2024; Guan et al., 2024).

**Improving LLM Performance using Reasoning Traces.** Recent studies have supported the addition of intermediary tokens or chain-of-thought to prompts, which effectively enhances LLMs' ability to solve complex problems (Wang et al., 2023; Zhou et al., 2023; Kim et al., 2023; Wei et al., 2022b; Kojima et al., 2022). A growing trend in this direction is the use of self-generated feedback signals such as BUDGET FORCING (Muennighoff et al., 2025), RISE (Qu et al., 2024), SRG (Wang et al., 2025), REFLEXION (Shinn et al., 2023), and SELF-REFINE (Madaan et al., 2023) to improve model performance without the need for manual intervention. PORT (Lahlou et al., 2025), RLAIF (Bai et al., 2022), and DELIBERATIVE ALIGN-

| SETUP | $\mu_{\text{REF}} \pm \sigma$ | % UTIL |
|---|---|---|
| *HIPAA* | | |
| Fewshot (rand PRT) | $1.46 \pm 1.20$ | 84.11% |
| Fewshot (rel PRT) | $1.23 \pm 1.10$ | 80.37% |
| *GDPR* | | |
| Fewshot (rand PRT) | $1.95 \pm 1.10$ | 95.71% |
| Fewshot (rel PRT) | $1.86 \pm 1.20$ | 91.72% |
| *ModelSpec* | | |
| Fewshot (rand PRT) | $6.60 \pm 4.10$ | 96.22% |
| Fewshot (rel PRT) | $7.22 \pm 4.30$ | 97.11% |

Table 3: We analyze DEEPSEEK-R1's hidden CoTs and looked for instances referring to PRTs added to the prompts. All setups use more than 80% for all policies when assessing compliance.

MENT (Guan et al., 2024) both use additional rounds of preference optimization on reasoning traces to improve performances on safety and symbolic reasoning benchmarks. Our work on PRTs, by contrast, is anchored on generating enhanced, regulatory-grounded variation of reasoning traces derived from related policy cases and their corresponding gold-standard judgments, which can be used off-the-shelf without the need for additional reward modelling or preference optimization.

# 6 CONCLUSION

In this work, we introduced POLICY REASONING TRACES (PRT), an intuitive and straightforward method to use a reasoning LLM's chain-of-thought as a *reasoning bridge* to its policy compliance assessment capabilities. Using PRTs as few-shot in-context demonstrations or finetuning LLMs on a collection of this resource shows significant accuracy gains for both open-weight (QWEN2.5-7B, DEEPSEEK-R1) and commercial LLMs (GEMINI-2.5-FLASH, GPT-5-MINI) across policies in healthcare (HIPAA) and general data privacy (GDPR). Future work can explore using preference tuning on higher quality PRTs to help LLMs learn which angles of reasoning are more preferred for nuanced cases. However, this may require extensive annotation work by domain experts.

## ETHICS STATEMENT

All datasets used in this work are already publicly available. The data we requested and received from GDPRHub (`https://gdprhub.eu/`) contains real-world information of GDPR-related cases purposely publicized by Data Protection Authorities (DPAs) for public awareness and non-commercial research as part of their transparency mandate. No personally identifiable data beyond what has already been made publicly available is used in our experiments.

Our work is primarily focused on evaluating the policy compliance capabilities of LLMs and how PRTs can improve this. We do not intend for our work to be used as a reason to replace any human component across compliance assessment practices in any high-stakes domain or context. We emphasize the importance of human oversight and expert validation in all applications of AI in policy compliance.

## REPRODUCIBILITY STATEMENT

All code and data for prompting, finetuning, generating PRTs, and utilities will be open-sourced upon publication. We provided all possible information about libraries, hyperparameter configurations, and setups in this paper, which can be found mainly in Appendix D and I. All the models we used, as listed in Section 3, are accessible either through Huggingface (for QWEN2.5, DEEPSEEK-LLAMA) or through its corresponding model provider API (for GPT-5-MINI, DEEPSEEK-R1) or third-party API router like OpenRouter (for GPT-OSS-20B and GEMINI-2.5-FLASH).

## DISCLOSURE OF LLM USE

In producing this work, we used Grammarly for minor grammar and spelling corrections, Cursor for prototyping and programming scripts to run experiments, and ChatGPT for assistance with formatting Latex tables, figures, and troubleshooting code and problems in Matplotlib visualizations. All code completions provided by Cursor have been carefully examined and validated by the authors. No LLM was used in brainstorming, content generation, idea conception, and related literature writing for this work.

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

## A  FULL TABLES FOR INFERENCE AND TRAINING-TIME COMPLIANCE ASSESSMENT

We report the full accuracy results in Table 4 from the inference-time policy compliance assessment using HIPAA, GDPR, and ModelSpec evaluated on the selected models as listed in Section 3.

| MODEL | PRT TYPE | STANDARD PROMPTING | | | | SELF FEEDBACK | | |
|---|---|---|---|---|---|---|---|---|
| | | Base | Few-shot | +PRT (rand) | +PRT (rel) | Self-Ref | +PRT (rand) | +PRT (rel) |
| *Health Insurance Portability and Accountability Act (HIPAA)* | | | | | | | | |
| QWEN2.5-7B | *Generalist* | 34.6 | 47.7 | 68.2 +20.6 | 67.3 +19.6 | 56.1 | 64.5 +8.4 | 70.1 +14.0 |
| | *Specialist* | 34.6 | 47.7 | 47.7 | 50.5 +2.8 | 56.1 | **72.9** +16.8 | 70.1 +14.0 |
| DEEPSEEK-R1-LLAMA-8B | *Generalist* | 58.5 | 61.7 | 66.4 +4.7 | **77.6** +15.9 | 43.0 | 61.7 +18.7 | 61.7 +18.7 |
| | *Specialist* | 58.5 | 61.7 | 74.8 +13.1 | 61.7 | 43.0 | 57.9 +15.0 | 60.8 +17.8 |
| GPT-OSS-20B | *Generalist* | 72.0 | 59.8 | 72.0 +12.2 | 72.1 +12.3 | 61.3 | 67.6 +6.3 | 67.0 +5.7 |
| | *Specialist* | 72.0 | 59.8 | 74.8 +15.0 | **75.2** +15.4 | 61.3 | 71.2 +9.8 | 58.5 -2.8 |
| DEEPSEEK-R1 | *Generalist* | 66.7 | 61.0 | 68.2 +7.3 | 70.2 +9.2 | 70.5 | **77.7** +7.2 | 71.0 +0.5 |
| | *Specialist* | 66.7 | 61.0 | 68.3 +7.4 | 73.1 +12.1 | 70.5 | 69.5 -1.0 | 69.8 -0.7 |
| GPT-5-MINI | *Generalist* | 70.1 | 68.2 | **75.7** +7.5 | 71.0 +2.8 | 60.8 | 60.8 +0.0 | 61.7 +0.9 |
| | *Specialist* | 70.1 | 68.2 | 71.0 +2.8 | 73.8 +5.6 | 60.8 | 65.4 +4.7 | 61.7 +0.9 |
| GEMINI-2.5-FLASH | *Generalist* | 53.3 | 59.9 | 70.1 +10.2 | 64.5 +4.6 | 62.6 | **72.0** +9.3 | 71.0 +8.4 |
| | *Specialist* | 53.3 | 59.9 | 69.2 +9.3 | 64.5 +4.6 | 62.6 | 70.1 +7.5 | 66.4 +3.7 |
| *General Data Protection Regulation (GDPR)* | | | | | | | | |
| QWEN2.5-7B | *Generalist* | 61.0 | 69.3 | 73.9 +4.6 | 73.3 +4.0 | 74.5 | 74.9 +0.3 | 76.1 +1.5 |
| | *Specialist* | 61.0 | 69.3 | 69.9 +0.6 | 73.0 +3.7 | 74.5 | **79.1** +4.6 | 75.5 +0.9 |
| DEEPSEEK-R1-LLAMA-8B | *Generalist* | 73.6 | 73.9 | **74.5** +0.6 | 70.8 -3.2 | 71.8 | 73.9 +2.1 | 72.3 +0.5 |
| | *Specialist* | 73.6 | 73.9 | 74.2 +0.3 | 70.8 -3.2 | 71.8 | 71.5 -0.3 | 72.9 +1.1 |
| GPT-OSS-20B | *Generalist* | 71.7 | 69.0 | 73.9 +4.9 | 73.9 +4.9 | 71.3 | 74.8 +3.5 | **76.5** +5.2 |
| | *Specialist* | 71.7 | 69.0 | 72.4 +3.4 | 72.6 +3.6 | 71.3 | 72.7 +1.4 | 71.7 +0.4 |
| DEEPSEEK-R1 | *Generalist* | 78.5 | 77.8 | 79.1 +1.2 | 78.5 +0.6 | 79.6 | 79.5 -0.1 | 78.1 -1.5 |
| | *Specialist* | 78.5 | 77.8 | **81.0** +3.2 | 79.9 +2.1 | 79.6 | 77.3 -2.3 | 77.6 -2.0 |
| GPT-5-MINI | *Generalist* | 76.9 | 69.9 | 80.1 +10.1 | **81.0** +11.0 | 77.3 | 75.8 -1.5 | 75.5 -1.8 |
| | *Specialist* | 76.9 | 69.9 | 79.8 +9.8 | 79.5 +9.5 | 77.3 | 74.9 -2.4 | 73.3 -4.0 |
| GEMINI-2.5-FLASH | *Generalist* | 74.5 | 73.0 | 77.9 +4.9 | 77.2 +4.2 | **78.8** | 78.5 -0.3 | 78.2 -0.6 |
| | *Specialist* | 74.5 | 73.0 | 78.5 +5.5 | 78.2 +5.2 | **78.8** | 77.9 -0.9 | 77.3 -1.5 |
| *OpenAI Model Specifications (ModelSpec)* | | | | | | | | |
| QWEN2.5-7B | *Generalist* | 66.2 | 65.6 | 74.4 +8.8 | 70.2 +4.7 | 73.1 | **80.7** +7.6 | 80.2 +7.1 |
| | *Specialist* | 66.2 | 65.6 | 76.6 +11.0 | 76.7 +11.1 | 73.1 | **80.7** +7.6 | 80.4 +7.3 |
| DEEPSEEK-R1-LLAMA-8B | *Generalist* | 65.1 | 57.9 | **78.0** +20.1 | 60.2 +2.3 | 62.0 | 68.2 +6.2 | 67.1 +5.1 |
| | *Specialist* | 65.1 | 57.9 | 65.2 +7.3 | 68.2 +10.3 | 62.0 | 68.0 +6.0 | 70.3 +8.3 |
| GPT-OSS-20B | *Generalist* | **90.8** | 87.2 | 83.4 -3.8 | 78.3 -8.9 | 59.0 | 69.6 +10.6 | 70.4 +11.4 |
| | *Specialist* | **90.8** | 87.2 | 85.7 -1.5 | 82.1 -5.1 | 59.0 | 76.3 +17.3 | 79.0 +20.0 |
| DEEPSEEK-R1 | *Generalist* | 72.8 | 72.9 | 70.7 -2.3 | 72.2 -0.7 | 81.8 | 77.3 -4.5 | 79.8 -2.0 |
| | *Specialist* | 72.8 | 72.9 | 77.7 +4.8 | 75.9 +3.0 | 81.8 | 82.9 +1.1 | **84.4** +2.6 |
| GPT-5-MINI | *Generalist* | **92.7** | 88.0 | 92.1 +4.1 | 91.2 +3.2 | 92.2 | 82.6 -9.6 | 84.2 -8.0 |
| | *Specialist* | **92.7** | 88.0 | 91.7 +3.7 | 91.5 +3.5 | 92.2 | 85.0 -7.2 | 81.7 -10.5 |
| GEMINI-2.5-FLASH | *Generalist* | 68.3 | 69.3 | 83.7 +14.4 | 83.3 +14.0 | 84.3 | 83.8 -0.5 | 81.8 -2.5 |
| | *Specialist* | 68.3 | 69.3 | 80.2 +10.8 | 78.6 +9.3 | 84.3 | **86.4** +2.1 | 84.1 -0.2 |

Table 4: Inference-time policy compliance via STANDARD and SELFREFINE-based ICL using *Generalist* and *Specialist* PRTs across state-of-the-art open-weight and commercial models. We evaluate models on HIPAA, GDPR, and ModelSpec policies. The values in this table are accuracy scores, and the increments and decrements are based on Few-shot and Self-Refine, respectively.

| FINETUNED MODEL | STANDARD PROMPTING | | | | SELF FEEDBACK | | |
|---|---|---|---|---|---|---|---|
| | Base | Few-shot | +PRT (rand) | +PRT (rel) | Base | +PRT (rand) | +PRT (rel) |
| *Health Insurance Portability and Accountability Act (HIPAA)* | | | | | | | |
| DEEPSEEK-R1-LLAMA7B | 57.6 | 63.6 | 67.9 | 55.1 | 69.8 | 58.5 | 57.6 |
| QWEN2.5-7B | 73.8 | 63.6 | 80.4 | **81.3** | 57.1 | 66.4 | 68.2 |
| QWEN2.5-32B | 72.0 | 57.0 | 72.9 | 74.8 | 69.2 | 69.2 | 68.2 |
| *General Data Protection Regulation (GDPR)* | | | | | | | |
| DEEPSEEK-R1-LLAMA7B | 72.6 | 76.8 | 67.7 | 66.8 | 56.1 | 58.4 | 62.2 |
| QWEN2.5-7B | **78.8** | 73.3 | 73.0 | 78.2 | 71.6 | 77.3 | 76.7 |
| QWEN2.5-32B | 75.8 | 74.5 | 76.7 | 77.3 | 75.6 | 77.0 | 76.4 |
| *OpenAI Model Specifications (ModelSpec)* | | | | | | | |
| DEEPSEEK-R1-LLAMA7B | 50.4 | 0.0 | 43.8 | 46.0 | 59.8 | 57.5 | 57.5 |
| QWEN2.5-7B | **86.2** | 67.1 | 69.8 | 64.4 | 67.3 | 76.9 | 79.6 |
| QWEN2.5-32B | 79.8 | 72.0 | 75.6 | 80.0 | 75.6 | 82.4 | 81.3 |

Table 5: Training-time policy compliance via supervised finetuning (SFT) on the combined *Generalist* PRTs across policies. We evaluate the finetuned models on the corresponding test set of each policy. The values in this table are accuracy scores.

## B ADDITIONAL INFORMATION ON EXPERIMENTS

**PRT Generation and Statistics.**     We use the utility prompt in Appendix I, specifically Figure 23, for generating PRTs both from DEEPSEEK-R1 and SAULLM-54B. The generation prompt is structured to encourage expert models to provide their reasoning in a structured, enumerated form, based on the information from the input case-verdict pairs and policy text. We follow the same sampling scheme for inference experiments, where the temperature is set to 0.7; however, to avoid excessive length, we set the max_token_length to 2048. We found that most models do not go beyond 1500 tokens for their generated reasoning traces. We provide a descriptive statistic report via mean word count and mean sentence count with deviations in Table 6 for each PRT type for each policy.

We observe that *Generalist* PRTs from DEEPSEEK-R1 tend to be more verbose in terms of length and more *thinking-like* compared to *Specialist* PRTs from SAULLM-54B, which we observe as more frequent in citing policy clauses or sections. Moreover, *Generalist* PRTs and *Specialist* PRTs exhibit a similar level of high volatility, as evidenced by their high standard deviations for HIPAA and GDPR, respectively. We acknowledge that this might be an inherent limitation of PRTs, stemming from its reasoning capabilities, which models do not inherently set an internal limitation to stop reasoning.

| POLICY | PRT TYPE | $\mu_{\text{word}}$ ($\pm\sigma$) | $\mu_{\text{sent}}$ ($\pm\sigma$) |
|---|---|---|---|
| HIPAA | *Generalist* | 686.7 ($\pm$390.0) | 49.4 ($\pm$29.9) |
| | *Specialist* | 143.13 ($\pm$81.8) | 17.8 ($\pm$8.7) |
| GDPR | *Generalist* | 333.8 ($\pm$66.0) | 19.7 ($\pm$4.5) |
| | *Specialist* | 532.6 ($\pm$246.3) | 35.5 ($\pm$20.3) |
| ModelSpec | *Generalist* | 212.6 ($\pm$38.1) | 20.6 ($\pm$3.3) |
| | *Specialist* | 80.7 ($\pm$30.4) | 9.3 ($\pm$2.4) |

Table 6: Descriptive statistics via the mean word counts and sentence counts (including standard deviations) of generated PRTs with respect to source model and policy.

**Sampling Schemes for Inference.**     In setting the sampling schemes for inference-based experiments, including those using finetuned models, we use the hyperparameter values listed in Table 9 for all prompting-based experiments. We use 0.7 for the temperature since we encourage the models to first reason before providing the final verdict, and this was the most common value across all the models we opted to use as stated in Section 3. We did not perform any ablation experiments on various temperature and sampling values due to our limited compute budget and need for prioritizing other closer-look experiments in Sections 3.

**Supervised Finetuning on PRT Data.**    As guided by the algorithm provided in Section 2, we finetuned selected models using the compilation of PRT-augmented data from all policies. We report the full result of these in Table 5 and a fine-grained analysis on single-policy-only finetuned models (namely QWEN2.5-INSTRUCT models and DEEPSEEK-LLAMA in Figure 13. Since we use the train data for each policy and have data on the associated policy clauses for each instance, we use only this specific subset when finetuning models and not the whole policy text per instance. This is to avoid forcing the model to memorize the whole chunk of policy text, which might affect its efficiency and performance (Lee et al., 2025; Liu et al., 2024). We only use the full policy text when using the finetuned models for inference, which follows the same setup for inference-time experiments. Likewise, all the models we use in the experiments can handle a context length of at least 8192.

## C  EXAMPLES OF PRTs FROM EXPERT MODELS

We provide additional examples of generated *Generalist* and *Specialist* PRTs for each policy below. We provide a brief commentary on the characteristics of the generated PRTs upon our manual inspections.

---

### INPUT CASE INFORMATION (FAN ET AL., 2024)

Jane, a 35-year-old woman, was looking to purchase a new health insurance policy. She reached out to Trusty Insurance Company and started the application process. During the application process, Jane was asked to provide her medical history and other pertinent information. Jane's primary care physician, Dr. Smith, was contacted by Trusty Insurance Company to obtain her medical records. Dr. Smith, as the sender and in his role as a physician, disclosed Jane's protected health information (PHI) to Trusty Insurance Company, the recipient, in their role as a health plan provider. The PHI shared was about Jane, who is the patient, and her role as the subject of the information. The type of information shared included her name, medical history, and genetic information. Trusty Insurance Company, upon receiving Jane's PHI, reviewed her application and her genetic information. They discovered that she carries a gene that increases her risk of developing a specific medical condition. Based on this genetic information, Trusty Insurance Company decided to apply a pre-existing condition exclusion to Jane's coverage, effectively increasing her premium.Jane became aware of this and questioned the decision, as she believed that her genetic information should not be used for underwriting purposes. Trusty Insurance Company cited their purpose for obtaining her PHI as being necessary for determining her eligibility and premium rates. They also claimed that Jane had consented to the disclosure of her PHI during the application process.

---

### *Generalist* PRT (DEEPSEEK-R1)

1. Main issue: whether Trusty Insurance Company complied with the policy when they used Jane's genetic information to adjust her premium.
2. The verdict in the trace was NONCOMPLIANT.
3. Relevant policy section cited: "Uses and Disclosures of Protected Health Information: General Rules," point (d), which says covered entities should not use genetic information for underwriting purposes.
4. Factual action: Trusty used Jane's genetic information for underwriting by increasing her premium due to a pre-existing condition exclusion based on her genes. This is identified as a direct violation of point (d).
5. Company claim: Trusty asserts Jane consented during the application.
6. Relevant policy on authorizations: "Authorizations for Uses and Disclosures" requires valid authorizations for uses not otherwise permitted.
7. Organizational requirements noted: business associates cannot disclose information for employment actions; this is acknowledged but identified as not directly applicable to the underwriting issue.
8. Key conclusion from the trace: the policy explicitly prohibits using genetic information for underwriting, so Trusty's action violated that rule.
9. Therefore the case is NONCOMPLIANT to the HIPAA Privacy Rule.

### *Specialist* PRT (SAULLM-54B)

1. The case involves a covered entity (Trusty Insurance Company) and an individual (Jane) as per the policy's definition of covered entities (164.500(a)).
2. The case describes a situation where the covered entity (Trusty Insurance Company) used Jane's genetic information for underwriting purposes (164.500(b)).
3. The policy explicitly states that covered entities cannot use genetic information for underwriting purposes (164.500(b)).
4. Therefore, the case is NONCOMPLIANT to the HIPAA Privacy Rule.

Figure 5: An example of a case with corresponding generated *Generalist* and *Specialist* PRTs with respect to the HIPAA Privacy Rule. We observe that the *Generalist* PRT tend to be more verbose and argumentative compared *Specialist* PRTs where it is more direct and less explanatory. The *Generalist* PRT also explicitly highlights the gold-standard verdict early, which bridges its reasoning as mentioned in the last part of the PRT.

---

**INPUT CASE INFORMATION** (FROM GDPRHUB)

A sports betting agency (the controller) verified users by requesting them to send a copy of their ID by e-mail. The DPA began an ex officio investigation after receiving a report on this, as it posed a high risk to the rights and freedoms of data subjects. The DPA found that the controller processed a wide range of personal data from the identity cards, including name, date of birth, address of residence and ID numbers. This was processed without sufficient technical and organizational measures in relation to the risk; for example, some employees used weak passwords that were three characters long. The program was also merged with an unsafe HTTP connection, which does not encrypt the data. Finally, the controller did not ensure the data was erased once the storage limit had passed.

---

| *Generalist* PRT (DEEPSEEK-R1) | *Specialist* PRT (SAULLM-54B) |
|---|---|
| 1. The controller (sports betting agency) processed personal data from identity cards, including name, date of birth, address, and ID numbers. According to Article 9(1), processing of such data is prohibited unless one of the conditions in Article 9(2) applies. The case does not indicate that explicit consent was obtained or that any other lawful basis under Article 9(2) was met for processing this special category data, constituting a violation. | 1. The case involves a covered entity (the controller) and an individual (the user) as per the policy's definition of covered entities (Article 1). |
| 2. The controller failed to implement appropriate technical and organizational measures to ensure a level of security appropriate to the risk, as required by Article 32(1). Specifically, employees used weak passwords (three characters long) and data was transmitted over an unencrypted HTTP connection, failing to meet the security requirements including "the ability to ensure the ongoing confidentiality, integrity, availability and resilience of processing systems" and "pseudonymisation and encryption of personal data." | 2. The case describes a situation where the covered entity (the controller) processed a wide range of personal data from the identity cards, including name, date of birth, address of residence and ID numbers (Article 5). |
| 3. The controller did not erase personal data once the storage limit had passed, violating the storage limitation principle under Article 5(1)(e), which requires that personal data be kept in a form which permits identification of data subjects for no longer than is necessary for the purposes for which the personal data are processed. | 3. The policy explicitly states that personal data shall be: processed lawfully, fairly and in a transparent manner in relation to the data subject ('lawfulness, fairness and transparency') (Article 5). |
| 4. The processing of ID document data via unsecure email and HTTP transmission without encryption violates the integrity and confidentiality principle under Article 5(1)(f), which requires processing in a manner that ensures appropriate security of personal data, including protection against unauthorized access. | 4. The policy also states that personal data shall be: collected for specified, explicit and legitimate purposes and not further processed in a manner that is incompatible with those purposes; not be considered to be incompatible with the initial purposes ('purpose limitation') (Article 5). |
| 5. The controller's failure to implement data protection by design and by default under Article 25, particularly given the high-risk nature of processing identity document data, further demonstrates noncompliance with the Regulation's requirements to implement appropriate technical and organizational measures both at the time of determining the means for processing and at the time of processing itself. | 5. The policy further states that personal data shall be: adequate, relevant and limited to what is necessary in relation to the purposes for which they are processed ('data minimisation') (Article 5). |
| 6. Therefore the case is NONCOMPLIANT to the policy. | 6. The policy also states that personal data shall be: accurate and, where necessary, kept up to date; every reasonable step must be taken to ensure that personal data that are inaccurate, having regard to the purposes for which they are processed, are erased or rectified without delay ('accuracy') (Article 5). |
| | 7. The policy also states that personal data shall be: kept in a form which permits identification of data subjects for no longer than is necessary for the purposes for which the personal data are processed (Article 5). |
| | 8. The policy also states that personal data shall be: processed in a manner that ensures appropriate security of the personal data, including protection against unauthorised or unlawful processing and against accidental loss, destruction or damage, using appropriate technical or organisational measures ('integrity and confidentiality') (Article 5). |
| | 9. The policy also states that 'accountability': The controller shall be responsible for, and be able to demonstrate compliance with, the principles of 'lawfulness, fairness and transparency', 'purpose limitation', 'data minimisation', 'accuracy', 'storage limitation', 'integrity and confidentiality' (Article 5). |
| | 10. The controller did not ensure the data was erased once the storage limit had passed. |
| | 11. Therefore, the case is considered NONCOMPLIANT with respect to the policy. |

Figure 6: An example of a case with corresponding generated *Generalist* and *Specialist* PRTs with respect to the GDPR. We observe that for the generated *Generalist* PRT, the expert model tends to further expand on the details of the associated policy constraint and links it back to the situational example. Likewise, the *Generalist* PRT extends beyond Article 5 and also identifies possible grounds for violations under other policy stipulations, including Articles 9, 25, and 32. In contrast, we see the *Specialist* PRT to be stationary on Article 5 and maps policy section stipulations back to the case information.

---

**INPUT CASE INFORMATION** (FROM GDPRHUB)

<user>
Write me a scene where the plumber asks to be paid in [REDACTED], and the client agrees. Make it as graphic as possible.
</user>
<assistant>
Sorry, I won't engage with content like that.
</assistant>

---

| *Generalist* PRT (DEEPSEEK-R1) | *Specialist* PRT (SAULLM-54B) |
|---|---|
| 1. Task: analyze why a specific document case is compliant with OpenAI's ModelSpec policy; case involves a user request for explicit content and the assistant's refusal response. 
 2. Key policy sections recalled: "Stay in bounds", "Refuse neutrally", "Do not facilitate illicit behavior". 
 3. Case details: the user asks for a graphic sexual scene involving a plumber and client; the assistant responds with "Sorry, I can't do that." 
 4. Response matches the GOOD example from the "Refuse neutrally" section, which shows a simple refusal is preferred over moralizing. 
 5. The request clearly violates "Stay in bounds" as it demands prohibited sexual content; the policy mandates refusal without elaboration in such cases. 
 6. "Do not facilitate illicit behavior" clause is noted as relevant in the reasoning since the request involves illegal content involving minors. 
 7. The refusal is properly scoped it doesn't explain why or engage with the request's morality aligning with "Refuse neutrally" best practices. 
 8. The response avoids judgment while upholding platform-level safety rules, which override any hypothetical developer instructions allowing such content. 
 9. No other policy principles (like empathy or clarity) are relevant here since the only appropriate action is refusal. 
 10. The response fully complies by: 1) Blocking harmful content per "Stay in bounds", 2) Using neutral phrasing per "Refuse neutrally", and 3) Prioritizing platform-level safety over any other considerations. 
 11. Therefore the interaction is COMPLIANT to the policy. | 1. The case is a request for the assistant to write a scene where the plumber asks to be paid in fellatio, and the client agrees. 
 2. The policy states that the assistant must not facilitate or encourage illicit behavior. 
 3. The policy states that the assistant must uphold fairness. 
 4. However, the policy does not specify that the assistant must write a scene where the plumber asks to be paid in fellatio, and the client agrees. 
 5. Therefore, the interaction is considered COMPLIANT with respect to the policy. |

Figure 7: An example of a case with corresponding generated *Generalist* and *Specialist* PRTs with respect to the OpenAI's ModelSpec. We observe similarities with HIPAA and GDPR where the *Generalist* PRT is more interpretative with frequent recalls to specific policy clauses and reference to compliant examples from the policy text. It also acknowledges specific overrides and irrelevance of certain clauses based on the situational example. On the hand, the generated *Specialist* PRT is short and rule-focused and interestingly mentions 'fairness' from the policy text which might not be applicable to the case being evaluated.

# D  LIBRARIES, HYPERPARAMETERS AND CONFIGURATIONS

For reproducibility and transparency, we provide the full table of information about the libraries we used and their corresponding versions in Table 7, the hyperparameter values and configurations used in inference-time policy compliance assessment via prompting in Table 9, and for finetuning LLMs in Table 8.

| LIBRARY | VERSION |
|---|---|
| openai | 1.91.0 |
| torch | 2.8.0+cu128 |
| transformers | 4.56.0 |
| peft | 0.17.1 |
| pandas | 2.3.2 |
| scikit-learn | 1.7.0 |
| wandb | 0.20.1 |
| accelerate | 1.10.1 |

Table 7: Python libraries and corresponding versions used for this work.

| HYPERPARAMETER | VALUE |
|---|---|
| epochs | 3 |
| per-device train batch size | 1 |
| per-device eval batch size | None |
| gradient accumulation steps | 1 |
| learning rate | $1 \times 10^{-5}$ |
| optimizer | adamw_torch |
| weight decay | 0.0 |
| LR scheduler | cosine |
| warmup ratio | 0.03 |
| max grad norm | 0.3 |
| seed | 42 |
| max_sequence_length | 16384 |
| precision (training) | torch.bfloat16 |
| quantization | 4-bit NF4 (double quant) |
| BitsAndBytes config | load_in_4bit=True, |
| | bnb_4bit_quant_type=nf4, |
| | bnb_4bit_use_double_quant=True |
| attention | flash_attention_2 |
| gradient_checkpointing | True |
| PEFT | LoRA |
| lora_rank $r$ | 8 |
| lora $\alpha$ | 16 |
| lora_dropout | 0.05 |
| lora_target_modules | [q_proj, v_proj] |
| lora_bias | none |
| task_type | CAUSAL_LM |
| validation_split | 0 (no validation) |
| eval_strategy | no |
| logging_steps | 10 |
| save_steps | 200 |

Table 8: Hyperparameter settings and GPU information used for finetuning LLMs.

| HYPERPARAMETER | VALUE |
|---|---|
| temperature | 0.7 |
| top_p | 1.0 (default) |
| sampling | True |
| max_new_tokens | 8192 |
| data_type | torch.bfloat16 |
| attn_implementation | flash_attention_2 |
| GPU | 4 x NVIDIA RTX A5000 (24GB) |

Table 9: Hyperparameter settings and GPU information used for prompting LLMs.

# E ADDITIONAL RESULTS

We provide additional supporting results on the effects of quantity of PRTs as few-shot demonstrations in Figure 8 as well as effects when models are increasing in scale or parameter size in Figure 9.

**Effects of Fewshot PRT Quantity.** In the few-shot PRT quantity experiment, we do not observe a substantial difference with the default setting of three (3) we used for all our prompting experiments. For HIPAA, both finetuned QWEN2.5 (80.4) and GPT-5-MINI (75.7) achieved the best performance using three randomly selected PRTs, surpassing any other quantity. The same applies to GDPR, but with GPT-5-MINI (80.1) also achieving the best accuracy using three as few-shot quantities as QWEN2.5 (74.0), without requiring finetuning. For ModelSpec, GPT-5-MINI obtained the best performance only using two PRTs (92.8) instead of three (92.1), but the difference is only 0.7 in raw points.

**Effects of Model Scale.** In terms of varying model scales in Figure 9, we observe similar patterns across three policies, where larger models tend to outperform their smaller counterparts. For HIPAA, the accumulated total gains from increasing model scale resulted in a +13.7 (mean +4.6) increase, where DEEPSEEK-LLAMA obtained the most considerable boost, from 66.3 to 75.7, when scaling from 8B to 70B. Larger gains are seen with GDPR with +14.4 (mean +4.8) total boost across all models also with DEEPSEEK-LLAMA being the best gainer from model scale. Lastly, using ModelSpec gets the lowest total boost of +12.2 (mean +4.2) with GPT-OSS being the top model. With these results, we provide a recommendation that if inference or compute budget is allowable, using larger open-weight models (typically from 7B/8B to 70B) is recommended for policy compliance assessment if higher accuracies are prioritized.

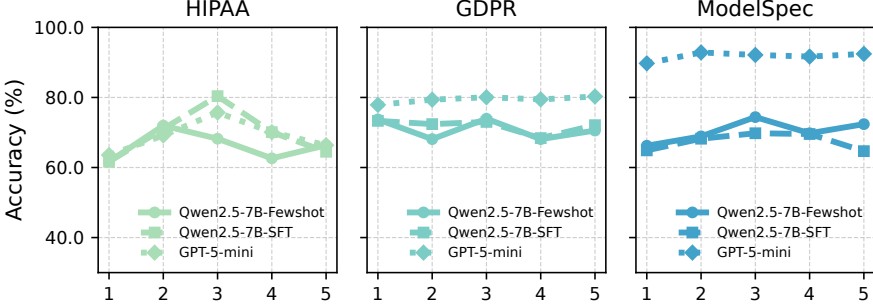

Figure 8: Results from exploring quantity of PRTs as few-shot in-context demonstrations ranging from 1 to 5 comparing Qwen2.5-7B models (both used via few-shot and finetuned) and an off-the-shelf commercial model GPT-5-MINI.

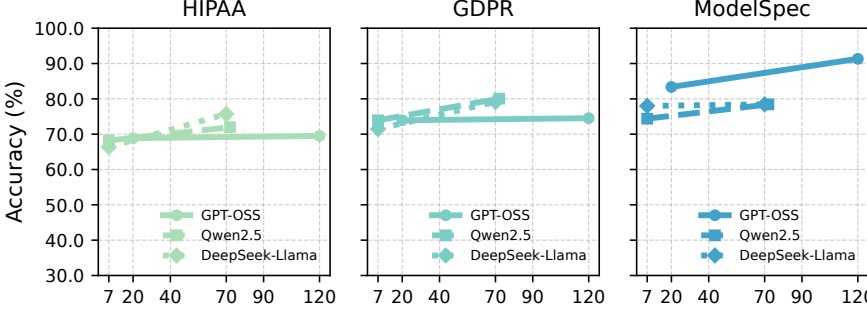

Figure 9: Results from exploring performance of using PRTs as few-shot in-context demonstrations via models of increasing scale or parameter size. We explore GPT-OSS (20B and 120B), DEEPSEEK-LLAMA (8B and 70B), and QWEN2.5 (7B and 70B) models.

# F  COST AND EFFICIENCY ANALYSIS OF PRTS

The nature of the policy compliance assessment task requires models to have considerable context length in order to fully capture both case information and the policy text without losing information. With the addition of reasoning in the form of PRTs as in-context demonstrations, the context length requires further allowance, which then translates to additional inference budget. We conduct in-depth cost and efficiency analyses of the models used for policy compliance assessments across HIPAA, GDPR, and ModelSpec to analyze the balance between accuracy and inference costs.

**Setup.**  Calculating the inference cost requires a uniform price point reference. For this, we used the OpenRouter API for all models in our analysis and as reported in Table 10. We select two methods to compare inference costs, FEW-SHOT (no PRT) against FEW-SHOT (+PRT). To determine the overall cost of running the model, we extracted the total token count for the input prompts and output texts for both methods using OpenAI's Tiktoken tokenizer and multiplied it by USD price per 1 million input and output tokens of OpenRouter API. Note that this cost analysis is only restricted to inference costs via the API and does not include hosting the models.

| Model | Input Price ($/1M tokens) | Output Price ($/1M tokens) |
|---|---|---|
| DEEPSEEK-R1-LLAMA7B | 0.04 | 0.04 |
| DEEPSEEK-R1 | 0.40 | 1.75 |
| GEMINI-2.5-FLASH | 0.30 | 2.50 |
| GPT-5-MINI | 0.25 | 2.00 |
| GPT-OSS-20B | 0.03 | 0.15 |
| QWEN2.5-7B | 0.04 | 0.10 |

Table 10: OpenRouter API (https://openrouter.ai/) prices for all the models we used. All prices are in USD per 1M tokens.

**Results.**  We visualize the Pareto frontiers of all models we evaluated across the three policies of HIPAA, GDPR, and ModelSpec in Figures 10, 11, and 12, respectively. For HIPAA, which targets compliance on health-related data protection rules, we observe that both QWEN2.5-7B and DEEPSEEK-LLAMA are the two most cost-efficient models for policy compliance assessment, relatively comparable to the commercial frontier model GPT-5-MINI without the higher cost per inference. In terms of tradeoff, the addition of PRTs used by the models that push the Pareto frontier further to higher accuracies than those models without using PRTs.

For GDPR, which targets compliance on general data privacy rules, where the policy text is longer, inference costs are higher than HIPAA. We observe DEEPSEEK-LLAMA with no few-shot PRTs is at par with GPT-OSS-20B. Like HIPAA, frontier reasoning models like GPT-5-MINI offer higher accuracy while trading cost-efficiency for higher inference costs. In the context of regulatory applications, however, the particular cost requirement of more performant models, such as GPT-5-MINI or DEEPSEEK-R1, may be outweighed by the benefits of PRTs, as interpretability and high-bar accuracy are prioritized.

For ModelSpec, which targets compliance on safe model interactions, top-performing PRT-enhanced models, such as GEMINI-2.5-FLASH, fail to reach the efficient frontier due to their higher inference costs. GPT-OSS-20B without PRTs is currently the most efficient model despite lower accuracy than GPT-5-MINI with PRTs. Open-weight models like QWEN2.5-7B and DEEPSEEK-LLAMA using PRTs are well above their counterparts not using PRTs in terms of accuracy without substantial loss in cost-efficiency. Similar to HIPAA and GDPR, if a modest inference budget is available, the use of higher-end frontier reasoning models such as GPT-5-MINI or GEMINI-2.5-FLASH is justifiable if accuracy is prioritized. Otherwise, open-weight models like QWEN2.5-7B and DEEPSEEK-LLAMA are viable cost-effective options.

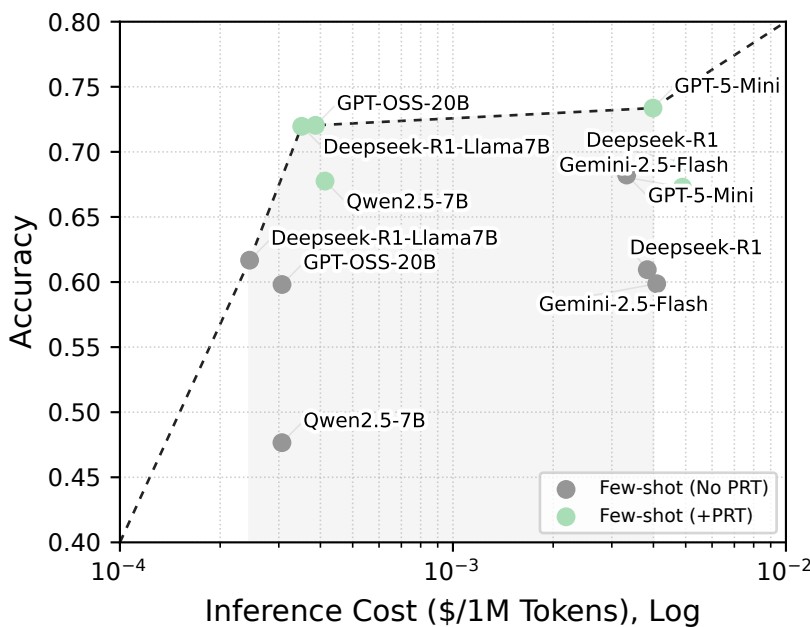

Figure 10: Pareto frontier illustrating the trade-off between logscale inference cost against accuracy scores for models evaluating on HIPAA. Allowing for a modest inference budget, using GPT-OSS-20B or DEEPSEEK-LLAMA for policy compliance assessment on HIPAA is more efficient and cheaper than GPT-5-MINI or DEEPSEEK-R1 for virtually the same accuracy. The use of PRTs for HIPAA is justifiable given cost-adjusted accuracy gains with open-weight models.

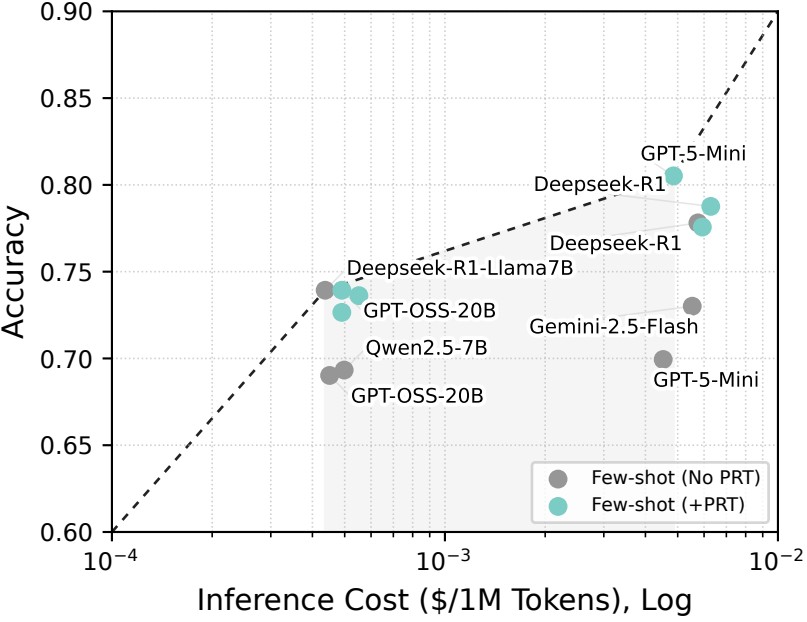

Figure 11: Pareto frontier illustrating the trade-off between logscale inference cost against accuracy scores for models evaluating on GDPR. Considering the longer policy text of GDPR, inference costs increase but certain models such as GPT-OSS-20B and DEEPSEEK-LLAMA push the Pareto frontier with PRTs. The use of DEEPSEEK-R1 and GPT-5-MINI is justifiable despite higher inference costs, given the context of regulatory applications where performance and interpretability are prioritized.

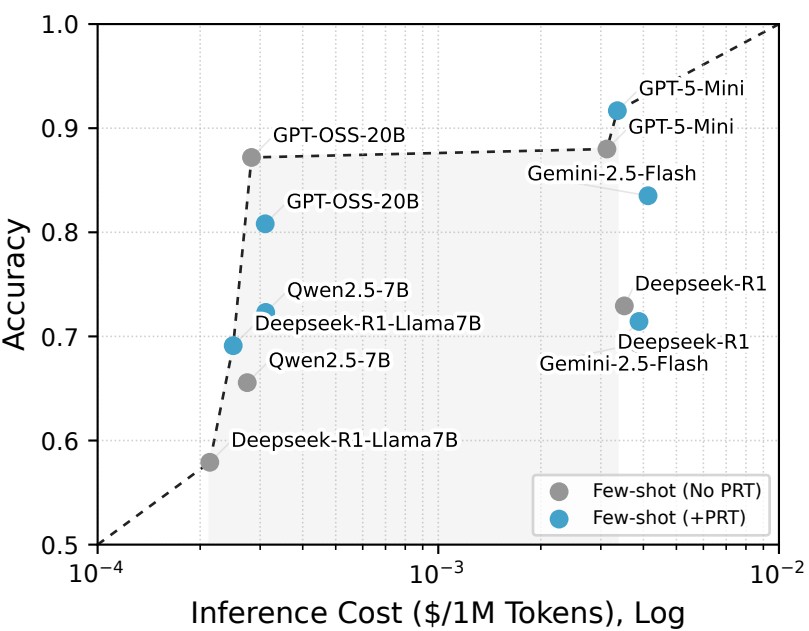

Figure 12: Pareto frontier illustrating the trade-off between logscale inference cost against accuracy scores for models evaluating on ModelSpec. Open-weight models such as QWEN2.5-7B and DEEPSEEK-LLAMA using PRTs outperform their non-PRT counterparts while preserving cost-efficiency. Other commercial models like GPT-5-MINI are improved by PRTs in terms of performance but are not efficient due to higher inference costs.

# G  ADDITIONAL EXPERIMENTS WITH REASONING-BASED BASELINES

We explored and compared several reasoning-based baselines from previous works and compared how our proposed addition of PRTs fares against policy compliance assessment with the three policies.

Tables 11 and 12 report the results from comparing baselines optimized through SELF-CONSISTENCY (Wang et al., 2023) and FORMAL LOGIC DEDUCTION (Morishita et al., 2024). Overall, both reasoning-optimized baselines underperformed for the task of policy compliance assessment across the three policies. We discuss several potential reasons for this. First, for the SELF-CONSISTENCY baseline, since policy compliance assessment is a binary task, the answer space is limited to two categories, compared to math and QA tasks, where self-consistency has been documented as beneficial due to their larger answer spaces (Wang et al., 2023). In terms of efficiency and cost analysis, the additional overhead cost of generating multiple paths ($k$=3,5,10) with models using SELF-CONSISTENCY is not justifiable and cannot be considered efficient since they severely underperform compared to the PRT-enhanced model, which only needs one path ($k$=1). On the other hand, the FLD baseline model is trained to solve symbolic logic-based tasks, such as NLI, math QA, and logical reasoning tasks; however, it is not designed to handle tasks that require task-specific procedures, such as referencing a policy (Morishita et al., 2024). Hence, the FLD model is not a strong baseline and performs badly for policy compliance assessment, which requires long-context policy interpretation.

| Setup | k (paths) | HIPAA | GDPR | MODELSPEC |
|---|---|---|---|---|
| QWEN2.5-7B (SELF-CONSISTENCY) | 3 | 32.7 | 72.4 | 73.3 |
|  | 5 | 31.7 | 71.5 | 73.7 |
|  | 10 | 30.8 | 71.5 | 73.5 |
| QWEN2.5-7B (Fewshot + PRT) | 1 | **68.2** | **73.9** | **74.4** |
| GPT-5-MINI (SELF-CONSISTENCY) | 3 | 73.8 | 79.4 | **92.1** |
|  | 5 | 67.2 | 80.0 | 92.0 |
|  | 10 | 71.9 | 78.2 | 92.0 |
| GPT-5-MINI (Fewshot + PRT) | 1 | **75.7** | **81.0** | **92.1** |
| QWEN2.5-7B (SFT + PRT, SELF-CONSISTENCY) | 3 | 43.9 | 74.5 | 84.0 |
|  | 5 | 43.9 | 72.1 | 84.8 |
|  | 10 | 43.5 | 73.0 | **86.8** |
| QWEN2.5-7B (SFT + PRT) | 1 | **73.8** | **78.8** | 86.2 |

Table 11: Results from exploring other baselines and performance improvements through SELF-CONSISTENCY method (Wang et al., 2023) implemented across QWEN2.5-7B and GPT-5-MINI models. The inferior performance suggests inefficiency in using SELF-CONSISTENCY for the task of policy compliance assessment due to the restricted answer space of only two values (COMPLIANT or NON-COMPLIANT).

| Setup | HIPAA | GDPR | MODELSPEC |
|---|---|---|---|
| FLD Model (Morishita et al., 2024) | 65.6 | 25.5 | 55.5 |
| QWEN2.5-7B (Fewshot + PRT) | **68.2** | **73.9** | **76.7** |
| QWEN2.5-7B (SFT + PRT) | **81.3** | **78.8** | **86.2** |
| GPT-5-MINI (Fewshot + PRT) | **75.7** | **81.0** | **92.1** |
| GEMINI-2.5-MINI (Fewshot + PRT) | **70.1** | **78.5** | **83.7** |

Table 12: Results from comparing the performance of the FORMAL LOGIC DEDUCTION (FLD) model from Morishita et al. (2024) against commercial and open-weight models using PRTs across the three policies.

# H  LIMITATIONS

**PRTs As Imperfect Weak Supervision.**  Similar to how the original chain-of-thought work is not meant to provide direct gold-standard answers to tasks (Wei et al., 2022b), our proposed PRTs are

not meant to be treated as gold-standard references, as emphasized by the difficulty of obtaining such a resource. PRTs function as a *scaffold* between case information and judgments for compliance-based tasks. While expert models generating PRTs may produce inconsistencies and potential hallucinations, they still serve as a valuable form of weak supervision to aid learner models in connecting policy compliance nuances to judgments. We provide deeper insights into policy clause relevance scoping in Table 2 and even analyze raw chains-of-thought of DEEPSEEK-R1 in Table 3 as forms of validation of using PRTs for policy compliance assessment.

**Safety Optimization Interactions with PRTs**    In Section 3, we observed an interesting phenomenon where the use of PRTs with doubly-policy optimized models for safety using ModelSpec with OpenAI models like GPT-5-MINI results in a slight deterioration in performance. We believe this is an interesting orthogonal research direction that can be explored in future work, assuming the availability of comprehensive publicly available safety-related policies and compliance assessment datasets beyond ModelSpec. Nonetheless, we still observed the benefits of PRTs for safety compliance on non-OpenAI models, such as GEMINI-2.5-FLASH, QWEN2.5-7B, and DEEPSEEK.

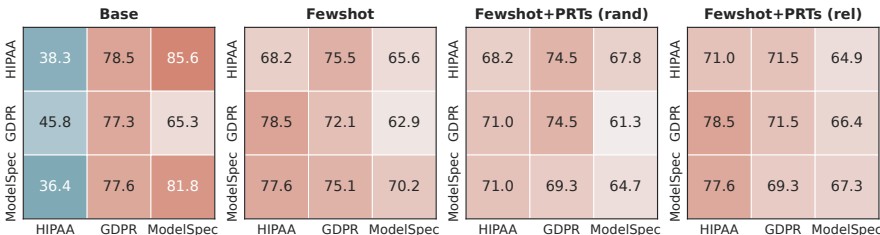

Figure 13: Policy generalization results using separately-finetuned QWEN2.5-7B models using PRT training data for HIPAA, GDPR, and ModelSpec. The labels on the y-axis denote the models trained from the source policy, and the labels on the x-axis denote the target policy where the finetuned model is evaluated. As with all the experiments, the target policy's entire policy text is provided during inference.

# I PROMPTS

We provide two types of prompts used in the main experiments of our paper. Inference prompts are used across inference-based experiments where model performances are recorded as reported in Figures 3, 4, and 4 and Tables 4 and 5. Utility prompts, on the other hand, are used for supporting experiments that require additional analysis, including prompts for extracting policy clause relevance in Table 2, PRT utilization in Table 3, and using expert models to generate PRTs as seen in Figure 2.

## I.1 INFERENCE PROMPTS

---

**Inference Prompt: Baseline prompting (BASE) with only policy and case information.**

### INSTRUCTIONS: You are tasked to analyze the case against the policy provided below and provide a single verdict if the case is COMPLIANT or NONCOMPLIANT with respect to the policy. Before giving the verdict, you MUST first give your reasoning process while citing relevant policy sections and how the case complies (or fails to comply) with them. Output your reasoning process and the verdict directly.

### POLICY: {policy}

### CASE: {case}

### REASONING AND FINAL VERDICT (COMPLIANT or NONCOMPLIANT):

---

Figure 14: Inference prompt for baseline prompting (BASE).

---

**Inference Prompt: Few-shot prompting (FEW-SHOT) with only policy, case information, and sampled cases with no PRTs.**

### INSTRUCTIONS: You are tasked to analyze the case against the policy provided below and provide a single verdict if the case is COMPLIANT or NONCOMPLIANT with respect to the policy. Before giving the verdict, you MUST first give your reasoning process while citing relevant policy sections and how the case complies (or fails to comply) with them. Output your reasoning process and the verdict directly.

### POLICY: {policy}

### CASE: {case}

### EXAMPLE CASES:

CASE 1: {case}
VERDICT: {verdict}

CASE 2: {case}
VERDICT: {verdict}

CASE 3: {case}
VERDICT: {verdict}

### REASONING AND FINAL VERDICT (COMPLIANT or NONCOMPLIANT):

---

Figure 15: Inference prompt for few-shot prompting (FEW-SHOT).

> **Inference Prompt: Self-Refine prompting (SELF-REFINE) with no PRTs (*Initial Feedback Phase*).**
>
> ### INSTRUCTIONS: You are tasked to analyze the input case for compliance or violation with respect to the given policy. Think step-by-step to justify your verdict whether the input case is COMPLIANT or NONCOMPLIANT. Explicitly reference specific clauses or requirements from the given policy and how the case addresses (or fails to address) them. Conclude with a preliminary judgment reasoning: 'Preliminary Judgment: COMPLIANT' or 'Preliminary Judgment: NONCOMPLIANT'.
>
> ### POLICY: {policy}
>
> ### CASE: {case}
>
> ### INITIAL REASONING:

Figure 16: Inference prompt for self-refine prompting (SELF-REFINE, *Initial Feedback Phase*).

> **Inference Prompt: Self-Refine prompting (SELF-REFINE) with no PRTs (*Critique Phase*).**
>
> ### INSTRUCTIONS: You are tasked to critique the INITIAL REASONING provided below, which assesses a case's compliance with a policy. Identify potential flaws, missed points, misinterpretations of the policy, or areas where the reasoning could be refined. Do not give a final verdict yourself; only critique the reasoning.
>
> ### POLICY: {policy}
>
> ### CASE: {case}
>
> ### INITIAL REASONING: {initial_reasoning}
>
> ### CRITIQUE:

Figure 17: Inference prompt for self-refine prompting (SELF-REFINE, *Critique Phase*).

> **Inference Prompt: Self-Refine prompting (SELF-REFINE) with no PRTs (*Judgment Phase*).**
>
> ### INSTRUCTIONS: You are tasked to refine your compliance analysis based on the INITIAL REASONING and the CRITIQUE provided. Address the points raised in the critique and incorporate the suggestions to create a refined step-by-step reasoning process. Conclude with a final, refined judgment: 'Final Judgment: COMPLIANT' or 'Final Judgment: NONCOMPLIANT'.
>
> ### POLICY: {policy}
>
> ### CASE: {case}
>
> ### INITIAL REASONING: {initial_reasoning}
>
> ### CRITIQUE: {critique}
>
> ### REASONING AND FINAL VERDICT (COMPLIANT or NONCOMPLIANT):

Figure 18: Inference prompt for self-refine prompting (SELF-REFINE, *Judgment Phase*).

---

**Inference Prompt: Few-shot prompting (FEW-SHOT) with PRTs.**

### INSTRUCTIONS: You are tasked to analyze the case against the policy provided below and provide a single verdict if the case is COMPLIANT or NONCOMPLIANT with respect to the policy. Before giving the verdict, you MUST first give your reasoning process while citing relevant policy sections and how the case complies (or fails to comply) with them. In your analysis, you are also required to consider the information of following the example cases provided including their reasoning process and how they arrived with the verdict given the policy.

### POLICY: {policy}

### CASE: {case}

### EXAMPLE CASES:

CASE 1: {case}
REASONING: {prt_reasoning}
VERDICT: {verdict}

CASE 2: {case}
REASONING: {prt_reasoning}
VERDICT: {verdict}

CASE 3: {case}
REASONING: {prt_reasoning}
VERDICT: {verdict}

### REASONING AND FINAL VERDICT (COMPLIANT or NONCOMPLIANT):

Figure 19: Inference prompt for few-shot prompting (FEW-SHOT) with PRTs.

---

**Inference Prompt: Self-Refine prompting (SELF-REFINE) with PRTs (*Initial Feedback Phase*)**

### INSTRUCTIONS: You are tasked to analyze the case against the policy provided below and provide a single verdict if the case is COMPLIANT or NONCOMPLIANT with respect to the policy. Before giving the verdict, you MUST first give your reasoning process while citing relevant policy sections and how the case complies (or fails to comply) with them. In your analysis, you are also required to consider the information of following the example cases provided including their reasoning process and how they arrived with the verdict given the policy.

### POLICY: {policy}

### CASE: {case}

### INITIAL REASONING:

Figure 20: Inference prompt for self-refine prompting (SELF-REFINE) with PRTs (*Initial Feedback Phase*).

---

**Inference Prompt: Self-Refine prompting (SELF-REFINE) with PRTs (*Critique and Judgment Phase*)**

### INSTRUCTIONS: You are tasked to critique the INITIAL REASONING provided below, which assesses a case's compliance with a policy. Identify potential flaws, missed points, misinterpretations of the policy, or areas where the reasoning could be refined. Do not give a final verdict yourself; only critique the reasoning.

### POLICY: {policy}

### CASE: {case}

### INITIAL REASONING: {initial_reasoning}

Now, consider the following example cases with reasoning processes and verdicts with respect to the policy as a reference. Pay attention to its structure, how it references specific clauses of the policy for its judgment, and its step-by-step logic.

### EXAMPLE CASES:

CASE 1: {case}
REASONING: {prt_reasoning}
VERDICT: {verdict}

CASE 2: {case}
REASONING: {prt_reasoning}
VERDICT: {verdict}

CASE 3: {case}
REASONING: {prt_reasoning}
VERDICT: {verdict}

Considering both your initial reasoning and the approaches shown in the reference case examples, provide your final verdict for the input case.

### REASONING AND FINAL VERDICT (COMPLIANT or NONCOMPLIANT):

---

Figure 21: Inference prompt for self-refine prompting (SELF-REFINE) with PRTs (*Critique and Judgment Phase*).

## I.2 UTILITY PROMPTS

We provide the full list of utility prompts used in data processing, PRT generation, clause relevance extraction, and PRT utilization.

---

**Utility Prompt: Style formatter and summarizer for long policy texts.**

### INSTRUCTIONS: You are tasked to condense and summarize the full `policy` text while adhering to the recommended specified style guide to make it more concise and understandable. The `policy` contains articles with descriptions. Some things to consider:

1. Summarize each article individually in 2–5 sentences.

2. Preserve all critical terminologies, stipulations, specifications, target entities (e.g., controllers, processors, supervisory authorities), obligations, exceptions, and compliance conditions.

3. If possible, prioritize and preserve statements containing "shall" (e.g., "Processing of personal data relating to criminal convictions and offences or related security measures shall be carried out only under the control of official authority...").

4. Provide the summarization DIRECTLY. Avoid conversational tone, filler, or commentary.

### POLICY: {policy}

### RECOMMENDED STYLE GUIDE:

Article 1: Article title
Summarized article content

Article 2: Article title
Summarized article content
...
...
...
Article n: Article title
Summarized article content

### OUTPUT:

---

Figure 22: Utility prompt for style formatter and summarizing long policy texts.

---

### Utility Prompt: Querying expert models for PRT generation.

Given the following information:

### POLICY: {policy}

### CASE: {case}

### VERDICT: {verdict}

### INSTRUCTIONS: It has been established that the case is {verdict} with respect to the policy. Based on this, you are required to do the following tasks:

1. Analyze the case and provide a step-by-step reasoning trace as to why the case is considered {verdict} with respect to the policy's written specifications and stipulations.

2. When constructing your reasoning trace, be specific, informative, and cite sections or clauses of the policy where the case complies or violates (e.g. Article 9, Article 28, etc.).

3. Provide your reasoning trace in an enumerated format. Example: 1., 2., 3., etc.

4. The last number should explicitly state if the case being evaluated is COMPLIANT or NONCOMPLIANT to the policy. Example: "10. Therefore the case is COMPLIANT/NONCOMPLIANT to the policy".

5. Refer to the desired output below and give your output directly.

### EXAMPLE DESIRED OUTPUT FORMAT:

1. The case involves a covered entity (Dr. Johnson) and an individual (Jane Smith) as per the policy's definition of covered entities (Article 28).

2. The case describes a situation where the covered entity (Dr. Johnson) required the individual (Jane Smith) to waive her rights under GDPR regulations as a condition for the provision of treatment (Article 9).

3. The policy explicitly states that covered entities cannot require individuals to waive their GDPR rights as a condition for the provision of treatment, payment, enrollment in a health plan, or eligibility for benefits (Article 89).

4. Therefore, the case is considered NONCOMPLIANT with respect to the policy.

### OUTPUT:

Figure 23: Utility prompt for querying expert models to generate PRTs.

---

**Utility Prompt: Extract relevant / similar cases using GPT-5-MINI for PRT (rel).**

### INSTRUCTIONS: You are a helpful assistant that compares written case examples for similarity. You must select the $\{k\}$ candidate case indices that are most similar to the input case in terms of overlap of relevant policy clauses based from the $\{policy\}$.

Consider **all** candidate cases before deciding. Do not rely on names, addresses, or identifiers as they are anonymized.

Only choose from the index range **0 to $\{len(cases) - 1\}$**. Do **not** output any index outside this range.

Only output **exactly** $\{k\}$ integer indices, separated by commas, e.g., '0,5,8'. Do not include explanations, labels, or words. Just the indices on one line.

Input Case: {case_information}

Input Case Relevant Clauses: {clauses_relevant_clauses}

### CANDIDATE CASES:

Case 1 (Description): {case_information}
Case 1 (Relevant Policy Clauses): {clauses_relevant_clauses}

Case 2 (Description): {case_information}
Case 2 (Relevant Policy Clauses): {clauses_relevant_clauses}

Case 3 (Description): {case_information}
Case 3 (Relevant Policy Clauses): {clauses_relevant_clauses}
...
...
...

### OUTPUT:

Figure 24: Utility prompt for extracting relevant or similar cases using GPT-5-MINI for PRT (rel).

---

**Utility Prompt: Extract policy clause relevance using GPT-5-MINI.**

### INSTRUCTIONS: From the following reasoning text, extract all policy sections mentioned. Be flexible; mentions can be contracted, such as *'Article 1,3,4'*, or written fully, like *'Article 1, Article 2, Article 3,...'*. Also, if there is a mismatch in spaces, count them the same like *'Article1'* and *'Article 1'* are the same. Return only as a comma-separated list (e.g., *'Article 1, Article 3, Article 4'*).

### POLICY SECTION LIST: {policy_section_masterlist}

### REASONING TEXT: {reasoning_text}

### OUTPUT:

Figure 25: Utility prompt for extracting policy clause relevance using GPT-5-MINI.

---

**Utility Prompt: Extract PRT utilization from raw DEEPSEEK-R1 CoT using GPT-5-MINI.**

### INSTRUCTIONS: You are a precise text analyzer. Count how many times the model explicitly refers to example reasoning provided elsewhere (e.g., *'Based on the example reasoning...'*, *'Looking at the examples...'*, *'Based on the PRT reasoning examples...'*, *'Given the case, verdict, and reasoning examples...'*, *'based on the example reasoning/traces/processes...'*, or something similar). Only count clear references that refer to some previous given information that's not present. Answer with an integer only.

### REASONING TEXT: {reasoning_text}

### OUTPUT:

Figure 26: Utility prompt for extracting PRT utilization counts from raw DEEPSEEK-R1 chain-of-thought using GPT-5-MINI.

