# OpenReview forum: "Scaling Policy Compliance Assessment in Language Models using Policy Reasoning Traces"
_ICLR.cc/2026/Conference — Submitted to ICLR 2026_

### Official Review · Reviewer_TNXz · 2025-10-30

**Soundness:** 2
**Presentation:** 2
**Contribution:** 2
**Rating:** 4
**Confidence:** 3

**Summary:**

This paper addresses the challenge of automated policy compliance assessment, a task that requires complex reasoning aligned with human-defined rules. The authors note that acquiring gold-standard expert reasoning traces for this task is prohibitively expensive. They propose a method called POLICY REASONING TRACES (PRT), which uses a powerful "expert" LLM to synthetically generate step-by-step reasoning traces, given only a case and its final compliance verdict (e.g., COMPLIANT/NONCOMPLIANT). These synthetically generated PRTs are then used as a "reasoning bridge"—either as few-shot in-context examples or as data for supervised fine-tuning—to improve the performance of a "learner" model. The authors demonstrate that this method improves accuracy and policy clause citation on the HIPAA and GDPR policy datasets.

**Strengths:**

1. Important Problem Domain: The paper tackles a practical and high-stakes problem. Automating policy compliance has clear real-world applications, and the challenge of aligning LLM reasoning with complex, human-authored policy documents is a significant research barrier.

2. Intuitive Core Idea: The central idea of using a more capable model to generate pseudo-expert reasoning, thereby avoiding the bottleneck of human expert annotation, is intuitive and a common-sense approach to data augmentation.

3. Strong Empirical Gains on Some Datasets: The method demonstrates clear and significant performance boosts over baseline models, particularly for open-weight models on the HIPAA dataset. The claim of achieving a new state-of-the-art on the GDPR dataset is also a notable empirical contribution.

**Weaknesses:**

1. Critical Unaddressed Risk: PRT Quality and Faithfulness: The entire method hinges on the assumption that the "expert" LLM generates high-quality, faithful reasoning traces. The paper provides no substantial validation of this. The expert model is given the gold verdict and asked to produce reasoning for it. This setup strongly encourages post-hoc rationalization, where the model may generate a plausible-sounding but incorrect reasoning path to justify the given answer. If the PRTs are flawed, unfaithful, or contain factual errors (e.g., wrong clause citations), the learner model is simply being trained on high-quality noise. This "garbage in, garbage out" risk is a fundamental methodological flaw that is not adequately addressed.

2. Questionable Novelty: The technique of using a strong model to generate synthetic data (especially reasoning steps or chains-of-thought) to train a weaker model is a well-established paradigm in NLP (e.g., data distillation, synthetic CoT generation). The paper fails to clearly articulate what makes "PRT" a novel method beyond applying this existing technique to the specific domain of policy compliance.

3. Confusing/Contradictory Results: The results on the ModelSpec dataset are concerning. The authors note that for models already optimized for this policy (OpenAI models), adding PRTs decreases performance. This finding, which is somewhat downplayed, severely undermines the method's generalizability. It suggests that these "expert-generated" PRTs may be of lower quality than the model's own internal reasoning, further supporting the concern in Weakness 1.

4. Superficial Analysis of PRT Sources: The paper generates PRTs from a "Generalist" (DeepSeek-R1) and "Specialist" (SAULLM) model but then states that the Generalist PRTs were "majority advantage" and used for subsequent experiments. This is counter-intuitive (why would a specialized legal model be worse?) and this finding is not explored. This feels like a missed opportunity for analysis and weakens the paper's contribution.

**Questions:**

1. PRT Quality Validation: Was any human evaluation conducted to measure the quality, faithfulness, and factual correctness of the generated PRTs? How do we know the "expert" model's reasoning for a verdict is the correct reasoning, and not just a plausible hallucination?

2. Forced Justification vs. Organic Reasoning: How does this method compare to simply prompting the expert model with standard Chain-of-Thought (e.g., "Let's think step-by-step") and having it generate both the reasoning and the verdict? By providing the gold verdict, aren't you forcing the model to justify a conclusion, which encourages rationalization over genuine reasoning?

3. ModelSpec Performance Drop: Could you elaborate on the performance degradation for OpenAI models on ModelSpec? Does this not imply that PRTs are only beneficial as a crutch for weaker models, and may actually be harmful (by introducing noise) when a model's existing reasoning capability is already strong?

4. Generalist vs. Specialist PRTs: Why do you hypothesize that the Generalist model's PRTs outperformed the Specialist (legal) model's PRTs? This seems to contradict the premise of using domain-specific expertise.

---

> ### Author Response · Authors · 2025-11-19
> **Response to Reviewer TNXz (1/2)**
>
> Thank you for the feedback, Reviewer TNXz. We are delighted that you found our work tackles an important and complex problem for LLMs; our proposed PRT approach to be intuitive, simple, yet effective; and our experiment results to be empirical and significant in terms of their contributions. Please find our response to your feedback below:
>
> > Critical Unaddressed Risk: PRT Quality and Faithfulness: The entire method hinges on the assumption that the "expert" LLM generates high-quality, faithful reasoning traces. The paper provides no substantial validation of this. The expert model is given the gold verdict and asked to produce reasoning for it.
>
> Thank you for this observation. We respectfully disagree that our proposed method with PRT has “no substantial validation”. In the PRT generation process, by providing verdict information, enables us to obtain structured, verdict-grounded reasoning patterns encoded in the PRTs, which ultimately benefit learner models. The process is not meant to provide information equal to gold-standard, but rather, a form of “silver-standard” supervision that is sufficient enough to guide learner models. The quality of the PRTs is actually positively reflected in our experiments, which serves as its validation. We observed that diverse learner models greatly benefit from PRTs across general policy compliance assessment (Section 4.1), model scale (Appendix E), and citing correct policy clauses (Section 4.3). We argue that these are strong proofs of the beneficial quality obtained by using PRTs compared to setups that do not use PRTs. We will make this very clear in Section 2 of the paper regarding PRT quality.
>
> > Questionable Novelty: The technique of using a strong model to generate synthetic data (especially reasoning steps or chains-of-thought) to train a weaker model is a well-established paradigm in NLP (e.g., data distillation, synthetic CoT generation).
>
> Thank you for this observation. Please see our general response on the many forms of novelty that our work introduces for the task of policy compliance assessment.
>
> > Confusing/Contradictory Results: The results on the ModelSpec dataset are concerning. The authors note that for models already optimized for this policy (OpenAI models), adding PRTs decreases performance.
>
> Thank you for this observation. We respectfully disagree that highlighting where our proposed PRT method interacts with a specific family of models (OpenAI’s GPT) internal model optimizations “severely undermines” the method’s effectiveness. As a form of good empirical science, it is our responsibility to highlight special cases where our proposed approaches may yield limited gains due to certain scenarios.
>
> To expand, the observed limited gains of PRTs are only exhibited in OpenAI’s GPT model family, as the ModelSpec policy was developed by OpenAI, and further internal optimizations were conducted using this policy to optimize the model family’s safety (Guan et al. 2024). This is where we believe the “doubly policy optimization” phenomenon happens. As a counterfactual, other commercial models like Google’s Gemini-2.5-Flash that do not conform to ModelSpec actually greatly benefit from the PRT method in terms of safety compliance assessment (Section 4.2)
>
> We actually find it interesting and not at all a weakness that our proposed PRT approach shows policy interactions with a specific family of models (OpenAI’s GPT). We believe this is a solid contribution that can encourage further work on policy compliance while observing optimization interactions. Even Reviewer BqWv on their feedback is encouraging us to further expand our observation of this paper.
>
> > Superficial Analysis of PRT Sources: The paper generates PRTs from a "Generalist" (DeepSeek-R1) and "Specialist" (SAULLM) model but then states that the Generalist PRTs were "majority advantage" and used for subsequent experiments. This is counter-intuitive (why would a specialized legal model be worse?) and this finding is not explored.
>
> Thank you for this observation. Our choice of selecting the PRTs from the Generalist model is only for closer-look experiments, but we provided a complete comparison for both Generalist and Specialist PRTs for the train and test-time compliance experiments. Moreover, we referred to SaulLM as a “Specialist” only because it is exclusively finetuned with legal domain data, and to differentiate this from the Generalist model. We do not assume that the Specialist model is equal with an actual legal expert. We will make this differentiation clear in the paper in Section 3.

---

> ### Author Response · Authors · 2025-11-19
> **Response to Reviewer TNXz (2/2)**
>
> > PRT Quality Validation: Was any human evaluation conducted to measure the quality, faithfulness, and factual correctness of the generated PRTs? How do we know the "expert" model's reasoning for a verdict is the correct reasoning, and not just a plausible hallucination?
>
> Thank you for this observation. Please see our earlier response regarding the validation and quality of PRTs, and how the PRT generation process inherently reduces hallucinations by grounding reasoning in verdict information.
>
> > Forced Justification vs. Organic Reasoning: How does this method compare to simply prompting the expert model with standard Chain-of-Thought (e.g., "Let's think step-by-step") and having it generate both the reasoning and the verdict?
>
> Thank you for this observation. As mentioned in the Introduction of the paper, one of our motivations is to propose a method that can help learner models perform policy compliance assessments even in the absence of gold-standard expert reasoning (this is the “genuine reasoning”). Hence, we developed the PRT approach to produce “silver-standard” reasoning chains that are grounded in verdict information, thereby avoiding hallucinations or deviations from the ground truth.
>
> In terms of comparison, we actually compared standard reasoning-based prompting that encourages learner models to reason first (similar to “Let’s think step-by-step) before providing a final verdict against PRT-based approaches for train and test-time policy experiment in Table 3, Appendix A.
>
> > ModelSpec Performance Drop: Could you elaborate on the performance degradation for OpenAI models on ModelSpec? Does this not imply that PRTs are only beneficial as a crutch for weaker models, and may actually be harmful (by introducing noise) when a model's existing reasoning capability is already strong?
>
> Thank you for this observation. Please see our earlier response regarding doubly optimized policies above. Additionally, we respectfully disagree that the method is only “beneficial as a crutch” to “weaker” models. Our experiments with PRTs involved diverse reasoning models, including Gemini-2.5-Flash, DeepSeek-R1-Distill, and Qwen2.5-7B, all of which have strong reasoning capabilities and exhibited strong performance gains for policy compliance assessments with ModelSpec using PRTs (Table 3, Appendix A).
>
> > Generalist vs. Specialist PRTs: Why do you hypothesize that the Generalist model's PRTs outperformed the Specialist (legal) model's PRTs? This seems to contradict the premise of using domain-specific expertise.
>
> Thank you for this observation. The premise of domain-specific expertise serves as an example to contextualize the nature of policy compliance assessment, rather than asserting that domain-specific models will always produce better reasoning traces. We hypothesize that the Generalist model generates more effective PRTs because it is trained on more diverse datasets, resulting in better generalization and reasoning quality. The Specialist model, as shown in the visual examples in the paper, tends to produce narrower, citation-heavy explanations that do not generalize as well to varied compliance scenarios, especially beyond the legal domain. We will clarify this point in the paper and expand the discussion on why the Generalist model’s reasoning style is more effective for both training- and test-time compliance performance in our setting.
>
> Thank you again, Reviewer TNXz. We hope our detailed responses address your comments and help improve your overall assessment of the paper.

---

> ### Author Response · Authors · 2025-11-24
>
> Thanks for your feedback, Reviewer TNXz! Please let us know if we were able to clarify your questions about the paper, especially regarding the precedent-based process (verdict information is included) and the various forms of novelty that PRT introduces, which we addressed in the global response. We hope it will help you in updating your overall assessment of the paper.
>
> Thank you very much.

---

### Official Review · Reviewer_BqWv · 2025-10-31

**Soundness:** 3
**Presentation:** 4
**Contribution:** 3
**Rating:** 6
**Confidence:** 4

**Summary:**

This paper addresses the critical challenge of enabling LLMs to perform complex policy compliance assessments, a task that typically requires expensive, expert-human-generated reasoning data. The authors introduce PRT, a novel and scalable method for creating "pseudo-expert" reasoning chains. The method works by prompting a frontier "expert" LLM with a case, a policy, and its gold-standard verdict to generate a detailed reasoning trace. These PRTs are then used as a "reasoning bridge" to improve "learner" LLMs, either as few-shot demonstrations for in-context learning or as a dataset for SFT.
The authors evaluate this method on three diverse policies (HIPAA, GDPR, and OpenAI's ModelSpec) and demonstrate that it significantly enhances the performance of both open-weight and commercial models, setting new SoTA results on the HIPAA and GDPR benchmarks. The paper also provides deeper analysis showing that PRTs improve a model's ability to cite correct policy clauses and can enable strong cross-policy generalization.

**Strengths:**

1. The core contribution, using an expert LLM to generate post-hoc reasoning traces, is a highly practical and scalable solution to the data bottleneck problem in specialized domains like legal and policy compliance.
2. The method achieves SoTA performance on the HIPAA and GDPR benchmarks, notably beating the prior SoTA on HIPAA. The performance lift for open-weight models is particularly significant, with accuracy boosts of 50-100% bringing them much closer to commercial model performance.
3. The paper successfully demonstrates the value of PRTs in two distinct paradigms: as off-the-shelf few-shot examples for ICL and as a high-quality dataset for SFT.
4. The authors go beyond simple accuracy. The analysis showing that PRT-trained models are better at citing the correct policy clauses (Table 1) and the demonstration of strong cross-policy generalization (Figure 13)  add significant depth and credibility to the method's value.

**Weaknesses:**

1. The PRTs are generated by providing the expert model with the gold-standard verdict (Compliant/Non-Compliant) in the prompt. This means the generated PRT is a post-hoc rationalization of a known answer, not a genuine, "from-scratch" reasoning process to find the answer. This methodological choice may create unnaturally perfect or biased reasoning traces that do not reflect the difficulty of the original task.
2. The authors candidly report that their method hurts performance on OpenAI models (gpt-5-mini, gpt-oss) for the ModelSpec policy. While their "doubly-optimized" hypothesis is plausible, this is a significant failure case that demonstrates the method is not universally applicable and can be detrimental when a model is already highly aligned.
3. The entire method hinges on the quality of the "expert" LLM (deepseek-R1 and SaulLM). The paper acknowledges these PRTs are "imperfect weak supervision" and may contain hallucinations, but it does not deeply analyze the impact of this imperfection. If the expert LLM is systematically wrong or biased in its reasoning, the learner models will simply inherit and scale this flawed logic.

**Questions:**

1. The paper's "internal" compliance approach should be contrasted with "external" guardian models. Could the authors discuss the trade-offs (e.g., latency, cost, and robustness) of their method versus using a dedicated, external gurad like LlamaGuard, WildGuard[1][2], which provides runtime moderation and recovery feedback?
2. Given the SoTA claims, would the authors consider evaluating their PRT-finetuned models on the new, large-scale DynaBench [3]? This would provide a more direct comparison to other SoTA models designed for similar general-purpose, dynamic policy enforcement.
3. Regarding the PRT generation: What was the rationale for providing the gold-standard verdict to the expert model? Have the authors experimented with generating PRTs without this "answer key"? How does the quality and correctness of the expert's reasoning change in that more realistic scenario?
4. Can the authors elaborate on the "doubly-optimized" failure case? Is it possible this is a style conflict (e.g., the deepseek-R1 reasoning style conflicts with the internal alignment of GPT models) rather than just "overthinking"? This finding seems critical to understanding the method's limitations.


[1] https://arxiv.org/pdf/2312.06674

[2] https://arxiv.org/abs/2406.18495

[3] https://arxiv.org/abs/2509.02563

---

> ### Author Response · Authors · 2025-11-19
> **Response to Reviewer BqWv (1/3)**
>
> Thank you for the feedback, Reviewer QPkF. We sincerely appreciate that you consider our work to be a highly practical and scalable solution for policy compliance assessment, given its highly significant performance boosts with open-weight and commercial LLMs. We also appreciate that you recognize our work’s novelty beyond technique-wise, especially in findings that our method allows LLMs to better scope correct policy clauses and exhibit strong cross-policy generalization.
>
> > The PRTs are generated by providing the expert model with the gold-standard verdict (Compliant/Non-Compliant) in the prompt. This means the generated PRT is a post-hoc rationalization of a known answer, not a genuine, "from-scratch" reasoning process to find the answer. This methodological choice may create unnaturally perfect or biased reasoning traces that do not reflect the difficulty of the original task.
>
> Thank you for this observation. Please see our general response on the rationale behind adding verdict information to PRT and how this is grounded in general legal procedures on assessing compliance with regulations and policies. We would also like to clarify that the verdict information is only available for the PRT generation process using the training data. The genuine “from-scratch” reasoning occurs every time the learner model is prompted to predict the correct verdict for each instance in the test set, using the collection of generated PRTs from the training data as few-shot in-context examples.
>
> > The authors candidly report that their method hurts performance on OpenAI models (gpt-5-mini, gpt-oss) for the ModelSpec policy. While their "doubly-optimized" hypothesis is plausible, this is a significant failure case that demonstrates the method is not universally applicable and can be detrimental when a model is already highly aligned.
>
> Thank you for appreciating our openness about the current limitations of our work! We believe this is our responsibility to showcase good science that will help contribute to advancing the field, especially for a crucial multidomain task such as policy compliance assessment. We also believe that there is no method or algorithm in machine learning tasks that works universally all the time, as there will always be cases where it might fail or provide limited gains.
>
> We can only use OpenAI’s recent GPT-based models as a case example for the doubly-optimized models since these are so far the only family of LLMs that have been optimized for a specific policy, which is ModelSpec. However, we still do not view this as a significant detriment to the contributions of our work, since not all models out there conform to OpenAI’s notion of safety as outlined in the ModelSpec. Nonetheless, we will add this to our paper in the Limitations section, where we will emphasize that there are cases where this approach is more applicable and cases where it might provide limited gains.
>
> > The entire method hinges on the quality of the "expert" LLM (deepseek-R1 and SaulLM). The paper acknowledges these PRTs are "imperfect weak supervision" and may contain hallucinations, but it does not deeply analyze the impact of this imperfection. If the expert LLM is systematically wrong or biased in its reasoning, the learner models will simply inherit and scale this flawed logic.
>
> Thank you for this observation. As with any study using synthetic data, we must acknowledge that the synthetic reasoning data generated via PRTs is not equivalent to gold-standard expert reasoning. However, the PRT generation process we propose ensures the production of sufficiently high-quality, grounded reasoning chains that can serve as a form of supervision. By providing the verdict information, we obtain structured, verdict-grounded reasoning patterns encoded in the PRTs, which ultimately benefit learner models. The quality of the PRTs is positively reflected in our experiments, where the diverse learner models we investigate greatly benefit from PRTs across model scale (Appendix E) and citing correct policy clauses (Section 4.3) compared to adding non-PRT setups.

---

> ### Author Response · Authors · 2025-11-19
> **Response to Reviewer BqWv (2/3)**
>
> > The paper's "internal" compliance approach should be contrasted with "external" guardian models. Could the authors discuss the trade-offs (e.g., latency, cost, and robustness) of their method versus using a dedicated, external guard like LlamaGuard, WildGuard[1][2], which provides runtime moderation and recovery feedback?
>
> Thank you for the suggestion! We agree that it is sensible to compare our finetuned models, especially with ModelSpec for safety, with existing guard models like LlamaGuard3 and WildGuard. We were able to successfully run our ModelSpec-based experiments to compare with LlamaGuard3 and WildGuard which are both guardian models used for safety assessment. Please see results below:
>
> | Setup                                   | ModelSpec |
> |-----------------------------------------|-----------|
> | Best Baseline, No PRT (GPT-5-Mini)      | 92.7      |
> | **Best Fewshot + PRT (Gemini-2.5-Flash)**   | **86.6**      |
> | **Best SFT + PRT (Qwen2.5-7B)**             | **86.2**      |
> | LlamaGuard3                             | 59.3      |
> | WildGuard                               | 45.1      |
>
> We are pleased that our best models enhanced by PRT through fewshot in-context learning and finetuning both outperform external guard models like LlamaGuard3 and WildGuard. Of course, OpenAI models like GPT-5-Mini take the top as this is a heavily optimized model on ModelSpec. However, *the fact that the use of PRTs can effectively boost a model's performance on this particular task and policy without additional optimizations is strong evidence of its generalization capability*.
>
> In terms of latency and cost, the fact that LlamaGuard3 and WildGuard were unable to outperform Qwen2.5-7B (which is the cheapest model at $0.4/1M token) in our experiments suggests that they would not be able to push the Pareto frontier. It would mean that using finetuning Qwen2.5-7B with PRT or even using a commercial model like Gemini-2.5-Flash would remain *justified* compared to use these external guard models for policy compliance assessment.
>
> > Given the SoTA claims, would the authors consider evaluating their PRT-finetuned models on the new, large-scale DynaBench [3]? This would provide a more direct comparison to other SoTA models designed for similar general-purpose, dynamic policy enforcement.
>
> Thank you for the suggestion! Evaluating the finetuned models on general policy-based benchmarks like DynaBench can help further look into the generalizability of our models. We ran the test split of DynaBench as used in the paper with our PRT-finetuned Qwen2.5 and distilled DeepSeek-Llama8B models and present the results below:
>
> | Setup                                   | DynaBench |
> |-----------------------------------------|-----------|
> | LlamaGuard3                             | 13.1      |
> | WildGuard                               | 20.9      |
> | NemoGuard                               | 23.7      |
> | ShieldGemma                             | 38.2      |
> | GuardReasoner-8B                        | 22.0      |
> | DynaGuard-8B                            | 72.5      |
> | **Qwen2.5-7B (SFT + PRT)**                  | **56.9**      |
> | **Qwen2.5-32B (SFT + PRT)**                 | **79.0**      |
> | **Deepseek-Llama3-8B (SFT + PRT)**          | **61.3**      |
>
> From the results, we are very pleased to share that our finetuned Qwen2.5 and distilled DeepSeek-Llama8B were able to generalize and outperform most guard models available and even DynaGuard itself. This is, again, *strong evidence of the generalization capabilities provided through fine-tuning models with PRTs*, in addition to our cross-policy generalization experiments in Section 4.2.

---

> ### Author Response · Authors · 2025-11-19
> **Response to Reviewer BqWv (3/3)**
>
> > Regarding the PRT generation: What was the rationale for providing the gold-standard verdict to the expert model? Have the authors experimented with generating PRTs without this "answer key"?
>
> Thank you for this observation. Please see our general response on the rationale behind adding verdict information to PRT and how this is grounded in general legal procedures on assessing compliance with regulations and policies. Without providing verdict information during PRT generation, the reasoning chains from the expert models will not be grounded (and may potentially drift toward a wrong conclusion), and thus cannot serve as a reasoning bridge for learner models.
>
> > Can the authors elaborate on the "doubly-optimized" failure case? Is it possible this is a style conflict (e.g., the deepseek-R1 reasoning style conflicts with the internal alignment of GPT models) rather than just "overthinking"?
>
> Thank you for this observation to help diagnose the interesting bottleneck with our method. Our observation of the doubly-optimized failure case refers to the fact that GPT models have already been heavily optimized via RL based on the ModelSpec policy, and the additional PRT signals from prompts may conflict with these internal preferences. We are leaning more towards attributing it to overthinking since similar observations from previous works (Gema et al., 2025; Sui et al., 2025) are complementary and strong. Our counterfactual that further supports this is also that Gemini-2.5-Flash, which is not optimized with ModelSpec, gains substantial performance boosts using PRT from DeepSeek-R1.
>
> We sincerely thank Reviewer BqWv for suggesting that we compare our PRT method with these additional experiments. We hope our detailed responses address your comments and help improve your overall assessment of the paper.

---

> ### Author Response · Authors · 2025-11-24
>
> Thanks for your feedback and suggestion to try other baselines, Reviewer BqWv! We are pleased to share that we are able to compare our finetuned PRT models with external guard models like LlamaGuard3 and WildGuard for ModelSpec compliance as well as evaluating it on DynaBench. We are happy to know that our proposed PRT method allowed models to further generalize on DynaBench while outperforming the other safety-optimized guard models. **This has been so far the most interesting result we found out, and we sincerely thank you for this suggestion, as this further strengthens PRT as a novel method**. The results are now included in the updated manuscript. We hope it will help you in updating your overall assessment of the paper.
>
> Thank you very much.

---

### Official Review · Reviewer_QPkF · 2025-10-31

**Soundness:** 3
**Presentation:** 3
**Contribution:** 2
**Rating:** 2
**Confidence:** 3

**Summary:**

The paper proposes an approach for improving policy compliance checking via augmented training data with  synthetic data generation. For each rule, associated compliance examples with gold verdict labels, the proposed approach use an expert reasoning LLM to generate additional synthetic data. Then the generated synthetic data was used to finetune other LLMs for improving policy compliance reasoning.

The proposed method was validated against vanilla LLM without finetuning on the the generated policy reasoning traces in a few benchmark datasets. The proposed approach outperformed vanilla LLMs thanks to the robustness of the finetuning on synthetic data.

**Strengths:**

This is an interesting application paper where it demonstrates the use of synthetic reasoning traces in improving LLMs reasoning functions in a specific domain.

The paper was well written and it is easy to follow.

Some of the results are interesting to the application domain.

**Weaknesses:**

On the novelty and related baselines: There are a lot of similar approach not in the policy compliance but in general trying to create synthetic data from rules and examples to teach LLM reasoning:

+ Self‑Consistency Improves Chain‑of‑Thought Reasoning in Language Models (Wang et al., 2023) — ICLR 2023 poster/crux: https://openreview.net/forum?id=1PL1NIMMrw : generate many reasoning traces, not specific to one traces and select traces based on consistency, I think this method is more advanced since it does not only simply generate a reasoning trace but multiple ones then re-selecting based on  consistency

+ Enhancing Reasoning Capabilities of LLMs via Principled Synthetic Logic Corpus https://arxiv.org/html/2411.12498v1: as stated in the paper: " to enhance LLMs’ reasoning capabilities by program-generated logical reasoning samples. We first establish principles for designing high-quality samples by integrating symbolic logic theory and previous empirical insights." This should be a baseline to compare to

+ Enhancing Logical Reasoning in Large Language Models through Graph-based Synthetic Data https://arxiv.org/html/2409.12437v1: focus on improving LLMs using synthetic graph-based reasoning data

 Given that the ideas of generating synthetic data to augment the training data is not novel, the technical contribution for the machine learning field is limited.

**Questions:**

Could you please compare your approaches to the mentioned related works I listed in the weaknesses?

---

> ### Author Response · Authors · 2025-11-19
> **Response to Reviewer QPkF (1/4)**
>
> Thank you for the feedback, Reviewer QPkF. We are delighted that you found of work to be interesting and effective in improving reasoning in LLMs for policy compliance assessment. It also means a lot learning that our paper is readable and well-written to the public. Please find our response to your feedback below:
>
> > There are a lot of similar approach not in the policy compliance but in general trying to create synthetic data from rules and examples to teach LLM reasoning.
>
> Thank you for suggesting that we compare our PRT method with the mentioned works. We would like to clarify that we already compared our PRT experiments with the *previous highest-performing state-of-the-art methods for policy compliance*. This would be GoldCoin by Fan et al (EMNLP 2024) for HIPAA and Deliberative Alignment by Guan et al (OpenAI 2024). While we acknowledge the reviewer’s suggestion to explore some of the more general works related to improving reasoning including Self-Consistency (Wang et al., 2023) or the Principled Synthetic Logic Corpus (Morishita et al., 2024), we need to clarify that these are not specifically designed for policy compliance assessment and would therefore be at a *technical disadvantage* compared to GoldCoin, Deliberative Alignment, and our proposed PRT method. Moreover, we would like to comment on the fit of the suggested works for comparison:
>
> **Self-Consistency (Wang et al., 2023)** - This work is designed to improve the robustness (or consistency) of model predictions by asking models to produce n number of paths and identify the most consistent answer. However, policy compliance assessment is a fundamental binary task, where the output is either 'Compliant' or 'Non-Compliant'. Hence, the advantage of Self-Consistency, mostly exhibited with math problems and QA-based reasoning problems, where the answer space is substantially larger and infinite, may not translate to better performance with policy compliance assessment, where the answer space is binary.
>
> **Principled Synthetic Logic Corpus (Morishita et al., 2024)** - This work is designed for short-context symbolic deduction, not long-context policy assessment and interpretation. The policies we use contain approximately 4,000-9,000 tokens, which is impractical to convert to short logic representations and may potentially lose policy clause information. Using a synthetic logic corpus in our task would not produce meaningful or comparable results, as it neither addresses long-context policy interpretation nor the specific areas we are studying, including performance on compliance assessment, scoping correct policy clauses, and persistence in the learner model’s raw reasoning.
>
> **Enhancing Logical Reasoning through Graph-based Synthetic Data (Zhou et al., 2024)** - This work explores the use of synthetic graph-based data specifically for inductive reasoning and spatial reasoning, which are not directly applicable to policy compliance interpretation and assessment. They used datasets such as CLUTRR, where inputs are short, templated narratives derived from synthetic graphs, and outputs involve predicting simple relational facts (e.g., family relations like brother and sisters). The code and data are not publicly available, and the documentation of the algorithmic implementation is very limited, as it is a short paper.

---

> ### Author Response · Authors · 2025-11-19
> **Response to Reviewer QPkF (2/4)**
>
> Please see a more structured comparison of related works in the table below:
>
> | Method                                      | Domain                          | Clause Link | Verdict Condition | Policy Context | Works on Inference-Time |
> |---------------------------------------------|----------------------------------|-------------|--------------------|----------------|---------------------------|
> | Self-Consistency (Wang et al., 2023)         | generic reasoning                | ✗           | ✗                  | ✗              | ✓                         |
> | Synthetic Logic Corpus (Morishita et al., 2024) | logical reasoning              | ✗           | ✗                  | ✗              | ✗                         |
> | Graph-based Synthetic Data (Zhou et al., 2024) | inductive and spatial reasoning | ✗           | ✗                  | ✗              | ✗                         |
> | Deliberative Alignment (Guan et al., 2024)  | policy compliance                | ✓           | ✓                  | ✓              | ✗                         |
> | GoldCoin (Fan et al., 2024)                 | policy compliance                | ✓           | ✓                  | ✓              | ✗                         |
> | PRT (Ours)                                   | policy compliance                | ✓           | ✓                  | ✓              | ✓                         |
>
>
> We will add this Table to the paper in the Appendix to further emphasize that the methods we compared with the PRT method are already the strongest state-of-the-art (which we outperform) and clearly see the differentiation and fit for the task of policy compliance assessment.
>
> > Given that the ideas of generating synthetic data to augment the training data is not novel, the technical contribution for the machine learning field is limited.
>
> Thank you for this observation. Please see the general response regarding the novelty of our study. For additional clarification, we believe the novel contributions of our proposed PRT method extend beyond the mere approach of using synthetic reasoning chains. We demonstrate that PRTs enable models to generalize across policies, facilitate the scoping of accurate policy clauses, and persist in the reasoning models' chain of thought. *Taken together, these model improvements open novel potential research directions in automating policy compliance assessment*.

---

> ### Author Response · Authors · 2025-11-20
> **Response to Reviewer QPkF (3/4)**
>
> We are pleased to share the results from the additional experiments on implementing Self-Consistency (Wang et al., 2023) as a baseline. We selected one open-weight model (Qwen2.5-7B) and one commercial model (GPT-5-mini) for the comparison, as well as experimented with few-shot PRT and supervised finetuning (SFT) using PRT data. In consideration of our compute budget, we explore a range of $k=3,5,10$ paths for self-consistency.
>
> | Setup                                     | $k$ (paths) | HIPAA | GDPR | ModelSpec |
> |-------------------------------------------|-----------|-------|-------|-----------|
> | Qwen2.5-7B (Self-Consistency)             | 3         | 32.7  | 72.4  | 73.3      |
> | Qwen2.5-7B (Self-Consistency)             | 5         | 31.7  | 71.5  | 73.7      |
> | Qwen2.5-7B (Self-Consistency)             | 10        | 30.8  | 71.5  | 73.5      |
> | **Qwen2.5-7B (Fewshot + PRT)**                | **1**        | **68.2**  | **73.9**  | **74.4**      |
>
> | Setup                                     | $k$ (paths) | HIPAA | GDPR | ModelSpec |
> |-------------------------------------------|-----------|-------|-------|-----------|
> | GPT-5-Mini (Self-Consistency)                | 3         | 73.8  | 79.4  | **92.1**      |
> | GPT-5-Mini (Self-Consistency)             | 5         | 67.2  | 80.0  | 92.0      |
> | GPT-5-Mini (Self-Consistency)             | 10        | 71.9  | 78.2  | 92.0      |
> | **GPT-5-Mini (Fewshot + PRT)**                | **1**         | **75.7**  | **81.0**  | **92.1**      |
>
> | Setup                                     | $k$ (paths) | HIPAA | GDPR | ModelSpec |
> |-------------------------------------------|-----------|-------|-------|-----------|
> | Qwen2.5-7B (SFT + PRT, Self-Consistency)  | 3         | 43.9  | 74.5  | 84.0      |
> | Qwen2.5-7B (SFT + PRT, Self-Consistency)  | 5         | 43.9  | 72.1  | 84.8      |
> | Qwen2.5-7B (SFT + PRT, Self-Consistency)  | 10        | 43.5  | 73.0  | **86.8**      |
> | **Qwen2.5-7B (SFT + PRT)**                    | **1**         | **73.8**  | **78.8**  | 86.2      |
>
> Results show that using PRT both as few-shot in-context demonstrations (test-time compliance) or through finetuning (train-time compliance) both substantially outperform baseline models using self-consistency. As mentioned in our critique of self-consistency from the previous response, since policy compliance assessment is a binary task, the answer space is limited to two categories compared to math and QA tasks, where self-consistency is documented to be beneficial due to their larger answer space. In terms of efficiency and cost analysis, the additional overhead cost of generating multiple $k=3,5,10$ paths with models using self-consistency *is not justifiable and cannot be considered efficient* since they severely underperform compared to PRT-enhanced model which only need one path ($k=1$).

---

> ### Author Response · Authors · 2025-11-22
> **Response to Reviewer QPkF (4/4)**
>
> We are pleased to share the results of additional experiments comparing the baseline FLD-enhanced model from "Enhancing Reasoning Capabilities of LLMs via Principled Synthetic Logic Corpus" (Morishita et al., 2024), as recommended by Reviewer QPkF. For the experiments, since the FLD-enhanced model is a finetuned model, we compare it with one open-weight model, Qwen2.5-7B using few-shot PRT, a finetuned version using PRT data, and two commercial models, GPT-5-Mini and Gemini-2.5-Mini.
>
> | Setup                                      | HIPAA | GDPR | ModelSpec |
> |--------------------------------------------|-------|-------|-----------|
> | FLD-enhanced Model (Morishita et al., 2024) | 65.6  | 25.5  | 55.5      |
> | **Qwen2.5-7B (Fewshot + PRT)**                 | **68.2**  | **73.9**  | **76.7**      |
> | **Qwen2.5-7B (SFT + PRT)**                     | **81.3**  | **78.8**  | **86.2**      |
> | **GPT-5-Mini (Fewshot + PRT)**                 | **75.7**  | **81.0**  | **92.1**      |
> | **Gemini-2.5-Mini (Fewshot + PRT)**            | **70.1**  | **78.5**  | **83.7**      |
>
>
> Results show that all PRT-enhanced models, whether using PRTs as few-shot in-context examples without finetuning and using PRTs as finetuning data, outperform the FLD-enhanced model proposed by Morishita et al (2024) by a substantial margin across all policies (HIPAA, GDPR, ModelSpec).
>
> As mentioned in our previous response, we expected this performance difference due to a *task mismatch* between the models. The FLD-enhanced model is trained to solve symbolic logic-based tasks, such as NLI, math QA, and logical reasoning tasks, but not tasks that require task-specific procedures, such as referencing a policy (see Section 6 of Morishita et al (2024)). Hence, the *FLD-enhanced model is not a strong baseline*. This explains why the model performs badly for policy compliance assessment, which requires long-context policy interpretation.
>
> We sincerely thank Reviewer QPkF for encouraging us to explore these previous works, which further shed light on the effectiveness of our PRT framework. We will include all additional experimental results in the paper. We hope this additional result helps improve your overall assessment of our work.

---

> ### Author Response · Authors · 2025-11-24
>
> Thanks for your feedback and suggestion to try other baselines, Reviewer QPkF! We are pleased to share that we are able to finish the additional baselines (Self-Consistency and FLD) you suggested to include and show that our PRT-based method is still the top approach for policy compliance assessment. The results are now included in the updated manuscript. We hope it will help you in updating your overall assessment of the paper.
>
> Thank you very much.

---

### Official Review · Reviewer_uHM2 · 2025-11-02

**Soundness:** 3
**Presentation:** 3
**Contribution:** 1
**Rating:** 4
**Confidence:** 4

**Summary:**

The paper explores the policy compliance assesments and improve LLM's policy compliance assesment capabilities by Policy Reasoning Traces, that are extracted from LLMs that can make reasonings such as DeepSeek-R1. The authors combines compliance, policy, and reasoning to create a  dataset that contains these triplets. Then, they use it to improve perforamances of LLMs which can be trained using, SFT or ICL. The goal is to automate the process of checking compliances whether suits for the given policies by LLMs. It is expected to apply stipulations

**Strengths:**

- The paper touches on an important problem and provides an applicable solution.

- The proposed methodology improves the performance of open-weight and commercial models.

**Weaknesses:**

- The provided strategy is not novel, since the paper uses prompting strategies that can also be applicable to the other methods in the literature. The most novel part about the proposed methodology is the use of PTR, which sometimes decreases the performance of the models in SelfRef or doesn't make a change in Few-shot learning

- The experiments contain less training and test data, e.g., 309 and 107, which may cause models to overfit if they are SFTed. The number is even lower in the third dataset.

- Does PTR contain the answer to the compliance? If so, the trained models would only learn to extract the answer that passes inside the PTR.

- Compliance statements and policy documents are typically long inputs for most open-weight LLMs, and adding multiple examples for few-shot prompting or incorporating self-feedback further increases the context length. Moreover, using PRTs for each example does not seem to be a feasible solution.

- SelfRef + PTR sometimes decreased the performance of some models e.g. , GPT5-Mini

**Questions:**

- What is the average length of the context for each strategy, few-shot learning, Few-shot learning + PTR, selfRef, SelfRef + PTR?

---

> ### Author Response · Authors · 2025-11-19
> **Response to Reviewer uHM2**
>
> Thank you for the feedback, Reviewer uHM2! We are delighted that you believe our work addresses the important task of policy compliance assessment, and our proposed PRT method is a viable and scalable solution that benefits both open-weight and commercial LLMs. Please find our response to your feedback below:
>
> > The provided strategy is not novel, since the paper uses prompting strategies that can also be applicable to the other methods in the literature. The most novel part about the proposed methodology is the use of PTR, which sometimes decreases the performance of the models in SelfRef or doesn't make a change in Few-shot learning.
>
> Thank you for this observation. We agree that PRTs are derived from prompting but we also believe the novel contributions of our proposed PRT method extend beyond the performance boosts. Please see our general response on the many forms of novelty that our work introduces. Aside from this, while our PRT framework allows models to empirically and significantly gain performance boosts compared to previous works, *we believe in good science and that we should also report scenarios where it faces challenges or does not provide any gains at all*, which can serve as areas for future work.
>
> > The experiments contain less training and test data, e.g., 309 and 107, which may cause models to overfit if they are SFTed.
>
> Thank you for pointing this out. The HIPAA train and test dataset which contains 309 and 107 instances respectively, was from the published work of Fan et. al (EMNLP 2024). Hence, it is expected and fair for us to use the same split to compare our method’s performance with theirs. We also used the same LoRA-based finetuning by the study for the training-time compliance experiments, which, based on our judgment, is conservative and not overly fitting the model. The finetuning configurations are found in Appendix D.
>
> > Does PTR contain the answer to compliance? If so, the trained models would only learn to extract the answer that passes inside the PTR.
>
> Thank you for this observation. Please see our general response on the rationale behind adding verdict information to PRT and how this is grounded in general legal procedures on assessing compliance with regulations and policies.
>
> Regarding the concern on answer extraction, in supervised finetuning, we condition the learner models to predict a single target verdict v given the case information, policy text, and PRT from the train set, which are all combined together as an input. The PRT content, including its concluding phrase, is never part of the prediction target, which follows the same setup as GoldCoin (Fan et al, EMNLP 2024). At inference time, the learner model *does not receive a PRT for the test case*. It only receives the PRT demonstrations exclusively from the train set via few-shot or no PRTs. Therefore, the learner model cannot copy or extract the verdict from any source and must generalize the reasoning patterns to produce a verdict. We will add this clarification to our paper.
>
> > Compliance statements and policy documents are typically long inputs for most open-weight LLMs, and adding multiple examples for few-shot prompting or incorporating self-feedback further increases the context length. Moreover, using PRTs for each example does not seem to be a feasible solution.
>
> Thank you for pointing this out. Yes, the reviewer is correct that policy documents are normally long, which makes policy compliance assessment a long context-dependent task and a true challenge for state-of-the-art models. Regarding the feasibility of PRTs, we respectfully disagree that using PRTs is not a feasible solution. Our thorough cost-efficiency analysis in Appendix F shows that using PRTs with select open and commercial models pushes the Pareto frontier for both HIPAA and GDPR policies, which means that the high accuracy gains obtained by these models can effectively *justify* their inference costs and are efficient.  Moreover, as the context length of state-of-the-art models is becoming longer and cheaper per 1M token, we believe the Pareto frontier will be pushed even further with future model releases that are even more capable of reasoning.
>
> Thank you again, Reviewer uHM2. We hope our detailed responses address your comments and help improve your overall assessment of the paper.

---

> ### Author Response · Authors · 2025-11-24
>
> Thanks for your feedback, Reviewer uHM2! Please let us know if we were able to clarify your questions about the paper, especially regarding the precedent-based process (verdict information is included) and with the many forms of novelty that PRTs introduce, which we addressed in the global response. We hope it will help you in updating your overall assessment of the paper.
>
> Thank you very much.

---

### Author Response · Authors · 2025-11-19
**General Response**

We thank the reviewers for their thorough and constructive feedback on our study. We appreciate that the reviewers identified the following contributions in our work:

- *“touches on an important problem and proposes a simple but highly practical and scalable solution in policy compliance”* (BqWv, uHM2, QPkF)
- *“shows strong and significant empirical gains and state-of-the-art for both open-weight and commercial models”* (uHM2, BqWv, TNXz)
- *“goes beyond accuracy and shows evidence through deeper ablations that add depth to the proposed method, including citing correct policy clauses and cross-policy generalization”* (BqWv, TNXz)

Please find our general response and reviewer-specific response to your comments below. We aim for these detailed responses to address your questions and improve your overall assessment of the paper.

### **On the various forms of novelty of the study**

Reviewer uHM2, QPkF, and TNXz inquired about the novelty of PRTs and how synthetic data used to improve learner models through teacher models already exists. While this is correct in a general sense (and can be said to any related synthetic data-related study), we believe our proposed PRT framework’s novelty comes in many different forms, especially for the challenging task of policy compliance, as described below:

First, **using PRTs enables test-time compliance assessment without additional finetuning**. With respect to previous policy compliance methods, Deliberative Alignment (Guan et al., 2024) and GoldCoin (Fan et al.) require additional fine-tuning and preference optimization for compliance assessment prompts to work. We believe the use of PRTs is a strong advantage, especially if domain experts are specifically looking to use models for compliance-based tasks that are too large (e.g., 70-120B) to finetune or optimize on their own.

Second, the proposed work is the **first to explore providing learner models with examples of grounded reasoning from start (summarizing case information) to finish (finalizing a verdict) through PRTs**. The PRT generation pipeline (Section 2) ensures that the generated reasoning of PRTs continuously refers to the given gold-standard verdict. This notion of using example cases with reasoning and verdict to judge new cases is also grounded on principles of regulation and compliance (see rationale below).

Third, our ablation studies show that **PRTs open new opportunities for further understanding compliance dynamics of LLMs**. We showed that the use of PRTs helps models generalize across policies, which may be impactful for new policies with limited data (Section 4.2). We also showed that the addition of PRTs allows learner models to identify the correct policy clause that matches the experts’ own identified clauses (Section 4.3), which is extremely crucial in legal and safety domains.

### **On the rationale behind adding verdict information in generating PRTs**

Reviewer uHM2 and BqWv inquired about the rationale and motivation behind adding the verdict information when generating PRTs. In Section 2.1, as part of our method’s motivation, we discussed that the precedent-based process, where courts obtain reasoning from case precedents (previous case information and decisions) before issuing a verdict, is the common process and often emulated in legal NLP works ([Yang et al., 2025](https://openreview.net/forum?id=HL5X5uX0RD#discussion); [Chen & Zhang, 2025](https://dl.acm.org/doi/full/10.1145/3715928.3737487); [Liu et al., 2025](https://arxiv.org/abs/2508.03080); [Fan et al., 2024](https://aclanthology.org/2024.emnlp-main.195/)). Hence, our PRT generation method, where we provide information to the expert model to generate reasoning from the train set that will be used for the evaluation of new cases from the test set, is **perfectly grounded on realistic policy compliance assessment procedures**. We emphasize the importance of emulating and grounding this real-world process in how models perform the task of policy compliance assessment.

Removing the verdict information from the PRT generation process can be done for the purpose of experimenting just to compare performance changes, but this action removes the information on which the expert models can ground their reasoning and loses similarity with actual policy compliance assessment, as discussed above.

### Update (11/24)
We have updated and uploaded a new version of the paper reflecting all the suggested clarifications and results from additional experiments suggested by the reviewers. We thank you for the constructive feedback in improving our work and hope you will consider this in updating your overall assessment of the paper.

---

### Meta-Review · Area_Chair_UR8R · 2025-12-27

**Summary:**

The paper proposes to improve policy compliance checking capability via data synthesis. By prompting a larger model to generate synthetic data to train the student model. As mentioned by the authors, one of the reviewers (TNXz's ) fully used AI-generated reviews, hence the judgment is mainly based on the remaining three reviews. All reviewers agree that the paper is clearly written and solid, but with limited novelty. The proposed method has been demonstrated to be useful in many other domains, and this paper mainly applies it to a rather specific domain. Considering that ICLR typically expects a higher technical contribution, I would recommend rejecting this paper.

**Reviewer Concerns:**

limited novelty: this paper mainly adapts a well-known technique to a rather specific domain.

**Reviewer Scores:**

I do not see clear evidence that the reviewers would like to change their scores.

---

### Decision · Program_Chairs · 2026-01-26

Reject